# Progressive water deficits during multiyear droughts in basins with long hydrological memory in Chile.

Camila Alvarez-Garreton[1,2], Juan Pablo Boisier[1,3], René Garreaud[1,3], Jan Seibert[4], Marc Vis[4]

[1] Center for Climate and Resilience Research (CR2, FONDAP 15110009), Santiago, Chile
[2] Department of Civil Engineering, Universidad de La Frontera, Temuco, Chile
[3] Department of Geophysics, Universidad de Chile, Santiago, Chile
[4] Department of Geography, University of Zurich, Zurich, Switzerland

*Correspondence to*: Camila Alvarez-Garreton (calvarezgarreton@gmail.com)

**Abstract.** A decade-long (2010-2020) period with precipitation deficits in central-south Chile (30-41ºS), the so-called megadrought (MD), has led to streamflow depletions of larger amplitude than expected from precipitation anomalies, indicating an intensification in drought propagation. We analysed the catchment characteristics and runoff mechanisms modulating such intensification by using the CAMELS-CL dataset and simulations from the HBV hydrological model. We compared annual precipitation-runoff (P-R) relationships before and during the MD across 106 basins with varying snow/rainfall regimes and identified those catchments where drought propagation was intensified. Our results show that catchments' hydrological memory -modulated by snow and groundwater- is a key control of drought propagation. Snow-dominated catchments (30-35ºS) feature larger groundwater contribution to streamflow than pluvial basins, which we relate to the infiltration of snowmelt over the Western Andean Front. This leads to longer memory in these basins, represented by a significative correlation between fall streamflow (when snow has already melted) and the precipitation from the preceding year. Hence, under persistent drought conditions, snow-dominated catchments accumulate the effects of precipitation deficits and progressively generate less water, compared with their historical behaviour, notably affecting central Chile, a region with limited water supply and which concentrates most of the country's population and water demands. Finally, we addressed a general question: what is worse, an extreme single year drought or a persistent moderate drought? In snow-dominated basins, where water provision strongly depends on both the current and previous precipitation seasons, an extreme drought induces larger absolute streamflow deficits, however persistent deficits induce a more intensified propagation of the meteorological drought. Hence, the worst scenario would be an extreme meteorological drought following consecutive years of precipitation below average, as occurred in 2019. In pluvial basins of southern Chile (35-41ºS), hydrologic memory is still an important factor, but water supply is more strongly dependant on the meteorological conditions of the current year, and therefore an extreme drought would have a higher impact on water supply than a persistent but moderate drought.

# 1 Introduction

Persistent climatic anomalies may alter catchment response to precipitation. Thus, catchment dynamics under unusually multiyear precipitation deficits might not be correctly predicted based on the interannual variability over the last decades. This applies even when past decades include severe, but shorter dry conditions (Saft et al., 2016a). In other words, stationarity as commonly assumed for streamflow projections under climate change might be an invalid assumption (Blöschl and Montanari, 2010), which poses challenges for achieving realistic structures and parameters in hydrological models (Duethmann et al., 2020; Fowler et al., 2016).

Non-stationary catchment response modulates hydrological functioning. This applies particularly to drought propagation, i.e., the process leading to soil moisture droughts and hydrological droughts (streamflow and groundwater deficits) under dry meteorological conditions (Van Loon et al., 2014). While meteorological droughts are mainly controlled by regional precipitation, soil moisture and hydrological droughts are also controlled by catchment characteristics. Therefore, under similar meteorological conditions, the severity of hydrological droughts can vary significantly within a climatic region (Van Lanen et al., 2013). Most drought-related impacts on, for instance, agriculture, ecosystems, energy, industry, drinking-water and recreation depend primarily on groundwater and streamflow deficits (Van Loon, 2015). Therefore, understanding the geographical variation in drought propagation provides critical information for drought-hazard adaptation and mitigation (Van Loon and Laaha, 2015). In addition to such spatial variability, non-stationary catchment responses to precipitation would lead to a temporal variation in drought propagation.

This temporal aspect is becoming increasingly important since many regions around the globe are experiencing unprecedented long dry spells due to climate and circulation changes, causing unforeseen impacts on water supply (e.g., Schewe et al., 2014). Recent evidence has shown that protracted droughts may propagate differently within the same catchment (i.e., same landscape characteristics and governing runoff mechanisms) under similar precipitation deficits and temperature anomalies than shorter dry events. For example, studies in south-eastern Australia have reported changes in catchment functioning (Fowler et al., 2018; Saft et al., 2015; Saft et al., 2016b; Yang et al., 2017) during the Millennium drought that took place for more than a decade (1997-2010). More recently, Garreaud et al. (2017) reported an unprecedented decrease in annual runoff during a multiyear drought in central-south Chile, the so-called megadrought (MD). The amplified response of streamflow to a drought signal may be due to variations of drainage density related to depleted groundwater levels within the catchment (Eltahir and Yeh, 1999; Van De Griend et al., 2002), a factor also emphasised by Saft et al. (2016b).

The MD experienced in Chile since 2010 (and continuing up to date) offers a great opportunity to understand the potential impacts of global changes on hydrology and water supply over wide ranges of hydro-climatic regions and landscape characteristics. The persistency and geographical extension of the MD have few analogues in the last millennia, and its causes have been partially attributed to anthropogenic climate change (Boisier et al., 2016, 2018; Garreaud et al., 2017, 2019). This uninterrupted sequence of years with precipitation deficits has impacted various sectors, including coastal

ecosystems (Masotti et al., 2018), natural vegetation (Arroyo et al., 2020; Garreaud et al., 2017), fire regimes (Gonzalez et al., 2018) and water supply (Muñoz et al., 2020).

To deepen the understanding of the impacts of persistent droughts on water supply, we explore the mechanisms causing the larger-than-expected hydrological deficits in central-south Chile during the MD. We complement previous analyses of the MD in Chile (Garreaud et al., 2017; Muñoz et al., 2020) by incorporating four more years to the MD period, and by focusing

on drought propagation over 106 catchments located between 30ºS and 41ºS. The conceptual framework of our analysis is based on the water balance within a catchment, where the water sourced from precipitation takes different flow pathways and is temporally retained in various stores. In this scheme, the composite of response times associated to the different physical mechanisms transferring and storing water through the basin is referred as hydrological memory (Fowler et al., 2020).

For a dry year within a long drought, we can distinguish three cases: i) stationary drought propagation, when the streamflow

deficits are similar to those observed in isolated years (single year drought) with similar precipitation deficits; ii) intensified drought propagation, when streamflow deficits are larger than those observed in years with similar precipitation deficits; and iii) attenuated drought propagation, when streamflow deficits are lower than those observed in years with similar precipitation deficits. Based on previous studies relating groundwater dynamics to non-stationary catchment response to droughts (Carey et al., 2010; Eltahir and Yeh, 1999; Fowler et al., 2020; Saft et al., 2016b) we hypothesise that in catchments

with longer hydrological memory (i.e., catchments where water is retained for longer time in different storages such as aquifers and snowpack), the propagation of drought during multiyear precipitation deficits is intensified (i.e., larger streamflow deficits than those observed in years with similar precipitation deficits), when compared to single dry years.

To test this hypothesis, we characterised the historical precipitation and streamflow deficits at the catchment scale and followed Saft et al. (2015) to evaluate annual precipitation-runoff (P-R) relationships and identified those catchments where

drought propagation during the MD was maintained, intensified or attenuated with respect to their historical behaviour. We analysed catchment memory from observed hydrometeorological data and from the hydrological processes simulated by a bucket-type model calibrated for each basin. We related catchment hydrological memory with shifts in P-R relationships during the MD, and with drought propagation for different types of drought, from extreme single year droughts to moderate but persistent droughts (including the MD). Finally, we addressed a general question with practical implications: what is

worse in terms of water supply, a single year with extreme precipitation deficits, or several consecutive years with moderate deficits?

## 2 Study region and data

The study area corresponds to central-south Chile, spanning 9 out of 16 administrative regions between 30ºS and 41ºS (Fig. 1). Hydrometeorological data was obtained from the CAMELS-CL dataset (Alvarez-Garreton et al., 2018), including

catchment-scale daily precipitation from the CR2MET precipitation product (DGA, 2017) and streamflow time series for the period April 1979 to March 2020. Following the precipitation, snowmelt and runoff seasonality, spanning from austral

autumn to the summer of the following calendar year, the hydrological year in the study region is defined to run from April to March.

Monthly streamflow values were computed when 15 or more days had valid data. For those months, a mean monthly value was computed from the available daily values and then aggregated into the total number of days within the month to get total monthly runoff. Subsequently, gaps in monthly streamflow time series were filled based on a procedure previously used for monthly precipitation data (Boisier et al., 2016). The method uses multivariate regression models, taking advantage of the streamflow co-variability among multiple gauging stations in the study region (within or across basins). Using this approach, data for a station can be filled if the missing data do not exceed 25% of the period. Each missing month is independently assessed based on an ensemble of multivariate models based on covariant records from other stations as predictors. A given linear model is used if it shows a predictive power (coefficient of determination, $r^2$, larger than 0.75), otherwise, the missing month is not filled. Annual streamflow values were computed for stations and years where all monthly values were observed or could be filled.

We computed catchment-scale solid to total precipitation fractions and daily estimates for snowmelt over the period April 1981 to March 2020 based on the ECMWF surface reanalysis ERA5-Land dataset, available at a spatial resolution of 9 km (Muñoz-Sabater, 2019).

As a basis for the hydrological modelling (Sect. 3.1.2), we computed hypsometric curves for each catchment based on ASTER GDEM (Tachikawa et al., 2011).

The CAMELS-CL dataset includes catchment characteristics such as topography, geology and hydrological signatures such as the baseflow index representing the slow catchment response to precipitation. The anthropic-related data provided include land cover for the year 2016, the location of reservoirs, and granted water used rights within the basins.

From the 327 catchments located between 30ºS and 41ºS (Alvarez-Garreton et al., 2018), we selected 106 for this study based on the following criteria: i) no catchments with reservoirs to exclude effects of dam operations in runoff observations (56 basins excluded), ii) only catchments with less than 10% of their areas covered by glaciers to exclude the effects of glacier contribution in the propagation of droughts (5 basins excluded), iii) only catchments with more than 30 years of streamflow observations (96 basins excluded), and iv) only catchments where the annual rainfall explained less than 50% of the variance in annual runoff ($r^2 < 0.5$ in P-R regressions from Sect. 3.2.2; 64 basins excluded).

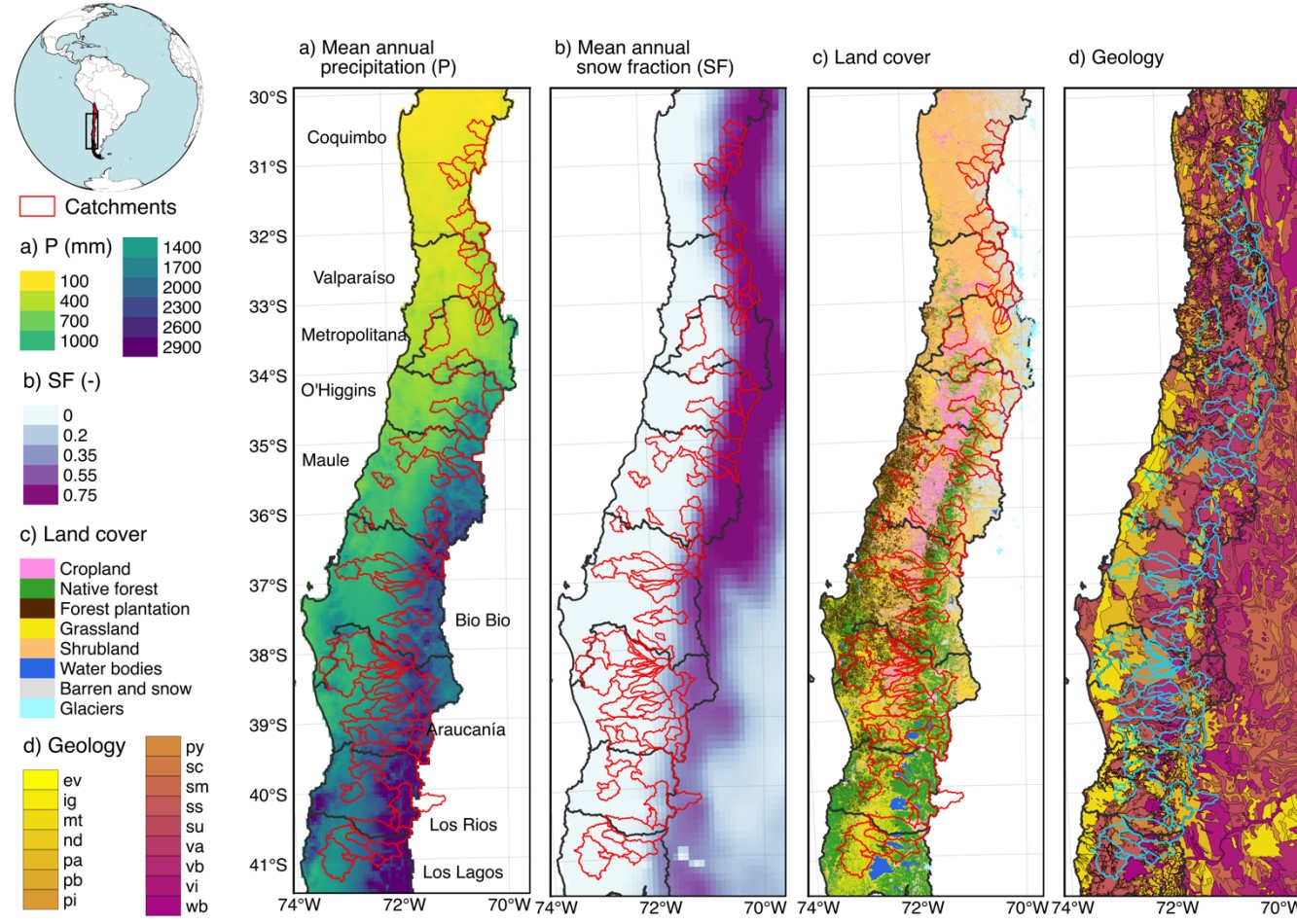

**Figure 1: Characteristics of the study domain and location of catchments. Panel a shows the mean annual precipitation from CR2MET gridded dataset and the 9 administrative regions covered by the study. Panel b shows the mean annual snow fraction from ERA5-L gridded dataset. Panel c shows the gridded land cover dataset from** Zhao et al., (2016)**. Panel d shows the geological classes from** Hartmann and Moosdorf, (2012)**, corresponding to evaporites (ev), ice and glaciers (ig), metamorphics (mt), no data (nd), acid plutonic rocks (pa), basic plutonic rocks (pb), intermediate plutonic rocks (pi), pyroclastics (py), carbonate sedimentary rocks (sc), mixed sedimentary rocks (sm), siliciclastic sedimentary rocks (ss), unconsolidated sediments (su), acid volcanic rocks (va), basic volcanic rocks (vb), intermediate volcanic rocks (vi), water bodies (wb).**

## 3 Methods

### 3.1 Hydrological analyses

### 3.1.1 Hydrological regimes

Given the latitudinal extent and terrain complexity, the study region features very different hydroclimate regimes (Fig. 1). Annual precipitation ranges from less than 100 mm to the north to near 3000 mm in the southern part. Precipitation also

increase substantially towards the west due to the orographic effect exerted by the Andes on the predominantly westerly atmospheric flow (Viale and Garreaud, 2014).

To characterise the different hydrological regimes of the study catchments, we classified them based on the hydro-climatic metrics summarised in Table 1, which represent the main seasonal hydro-climatic characteristics of a catchment. The classification was based on a k-means clustering algorithm (Lloyd, 1982) implemented with the H2O.ai library in Python (H2O.ai, 2020).

**Table 1: Hydro-meteorological basin features used for classification.**

| Variable | Description |
|---|---|
| $P_{A-S}$ ratio | Ratio of mean fall-winter precipitation (April to September) to mean annual precipitation (period April 1979 to March 2020) from CAMELS-CL dataset |
| $Q_{A-S}$ ratio | Ratio of mean fall-winter streamflow (April to September) to mean annual streamflow (period April 1979 to March 2020) from CAMELS-CL dataset |
| SF | Long-term snow fraction computed as the average of ERA5-L solid to total precipitation ratio for the period April 1981 to March 2020 |

**3.1.2 Modelling**

In addition to the analysis of the observations-based dataset from CAMELS-CL, we run the HBV model (Bergström, 1972; Lindström et al., 1997) to simulate streamflow and other fluxes for each of the 106 study catchments. With these simulations we seek to improve our understanding on runoff mechanisms from a process-based perspective, particularly regarding the role of snow and groundwater in runoff generation. The HBV is a bucket-type model that simulates the main hydrological 155 processes in a catchment through a number of routines. In the snow routine, snow accumulation and melt are simulated based on a simple degree-day approach. A variable fraction of all melted and rainfall water is retained in the soil depending on the current soil water level. The remaining part is transferred to the groundwater routine. In this routine, groundwater storage is represented by two boxes, an upper soil box representing faster groundwater release to total streamflow, and a lower soil box representing a slower groundwater release to total streamflow, both with linear outflows. Finally, the 160 simulated outflows from the groundwater stores are routed using a simple routing scheme, leading to the total streamflow. Besides streamflow, time series of a number of other fluxes and storages can be obtained from the model, such as actual evapotranspiration (ET), soil water storage, or the different streamflow components.

The HBV model has been implemented in several software packages. Here we used the version HBV light (Seibert and Vis, 2012). The model was calibrated using a genetic algorithm (Seibert, 2000) with parameter ranges similar to those suggested 165 earlier (e.g., Seibert and Vis, 2012). The 14 free parameters values were derived after 3500 model runs. For each catchment, 100 independent calibration trials were performed based on the non-parametric variation of Kling-Gupta efficiency, NPE (Pool et al., 2018), which resulted in ensembles with 100 parameter sets.

To characterise the slow groundwater contribution to runoff for each basin, we computed a groundwater index (GWI) as the mean annual outflow from the lower soil box (GW) normalised by the mean annual simulated streamflow. Note that GWI provides a measure of the groundwater flux simulated by HBV (i.e., not the groundwater storage). In contrast to the baseflow index provided in CAMELS-CL, which is computed from a low-pass filter applied to streamflow observations and thus represents the response timings to precipitation, the GWI represents timings and also the source of the water.

### 3.1.3 Quantifying hydrological memory

The hydrological memory of a catchment is the composite of response times associated with the physical mechanisms transferring and storing water through the basin (Fowler et al., 2020). Such response times have been qualitatively related to the presence of aquifers, lakes and snow (Van Loon and Van Lanen, 2012). Thus, there is no unique way to quantify hydrological memory. For example, catchment memory has been assessed based on soil moisture and groundwater dynamics (Agboma and Lye, 2015; Peters et al., 2006), on streamflow recession curves (Rodríguez-Iturbe and Valdes, 1979), on lag-correlations between soil moisture and other fluxes within the catchment (Orth and Seneviratne, 2013), and based on recovery times from droughts (Yang et al., 2017).

In this study, we assessed hydrological memory based on the following indices:

1) Seasonal streamflow memory, represented by the $r^2$ between fall-winter (April to September) precipitation of a certain year $t$ ($P_{A\text{-}S(t)}$) and the observed streamflow during the subsequent seasons: $Q_{OND(t)}$, $Q_{JFM(t)}$ and $Q_{AMJ(t+1)}$. These correlations represent the strength of the hydrological memory at 3, 6 and 9 months, respectively.

2) Seasonal GW memory, represented by the $r^2$ between $P_{A\text{-}S(t)}$ and the simulated GW during the subsequent seasons: $GW_{OND(t)}$, $GW_{JFM(t)}$ and $GW_{AMJ(t+1)}$. These correlations indicate hydrological memories of 3, 6 and 9 months, respectively.

3) Annual streamflow memory, represented by the $r^2$ between the residuals of the annual P-R regressions computed in Sect. 3.2.2 ($PR_{res(t)}$, i.e., annual streamflow not explained by the current annual precipitation) and the precipitation from the previous year ($P_{(t-1)}$). This index represents the information gained when the precipitation from the previous year is incorporated in the annual P-R relationships, thus indicating a hydrological memory beyond 12 months.

### 3.2 Drought propagation

### 3.2.1 Drought characteristics

Meteorological and hydrological droughts were characterised by the observed annual precipitation and streamflow anomalies at the catchment-scale, respectively. For each basin, the relative anomaly of streamflow ($Q_a'$) and annual precipitation ($P_a'$) in the hydrological year $t$ were computed as:

$$Q_a'(t) = (Q_a(t) - \overline{Q_a})/ \overline{Q_a}, \qquad (1)$$

$$P_a'(t) = (P_a(t) - \overline{P_a})/ \overline{P_a}, \qquad (2)$$

where $Q_a(t)$ and $P_a(t)$ are the annual streamflow observation and annual precipitation for the hydrological year $t$, respectively. $\overline{Q_a}$ and $\overline{P_a}$ are the mean annual streamflow and mean annual precipitation for the period April 1979 to March 2010 (i.e., excluding the MD), respectively. These relative anomalies are easy to interpret and commonly used in drought impact planning (Van Loon, 2015), but also have some limitations. Very large anomalies are obtained when the long-term mean is small. Furthermore, neither absolute nor relative deviations provide information about how unusual the anomalies are at specific locations. Therefore, we also computed the annual deviations of streamflow and precipitation normalised by the standard deviation of the annual time series for the period April 1979 to March 2010 (i.e., z-scores). Annual z-scores for streamflow ($Q_a{}^*$) and precipitation ($P_a{}^*$) were computed as follows:

$$Q_a{}^*(t) = (Q_a(t) - \overline{Q_a})/\,std(Q_a), \qquad (3)$$
$$P_a{}^*(t) = (P_a(t) - \overline{P_a})/\,std(P_a). \qquad (4)$$

To characterise the MD (April 2010 to March 2020) and to assess how unusual this 10-year period has been in comparison to previous decades, we computed the MD decadal means for each water flux (precipitation, observed streamflow and simulated ET) and compared them against historical decadal-mean values. To account for a large sample size, we computed 100 decadal mean values from random batches of ten annual values within the historical records.

### 3.2.2 Intensification of drought propagation during the MD

Stationarity in drought propagation during the MD was assessed by following the procedure suggested by Saft et al. (2015) to identify significant shifts in annual rainfall-runoff relationships over Australian catchments during the Millennium drought. Saft et al. (2015) showed that prolonged rainfall (liquid precipitation) deficits resulted in shifts in rainfall-runoff relationships at the catchment scale, and Saft et al. (2016b) related the shifts to catchment characteristics (aridity index and rainfall seasonality) and soil and groundwater dynamics. The physical mechanisms likely associated with these factors were discussed by Saft et al. (2016b), but not explicitly modelled.

In this study, we computed annual P-R relationships between annual precipitation (solid and liquid) and runoff time series for each catchment, and performed a global test to validate linear model assumptions with the R-package gvlma (Peña and Slate, 2006). From the 170 catchments fulfilling the criteria i) to iii) described in Sect. 2, we selected 106 where the linear assumptions in P-R relationships were fulfilled and where the annual rainfall explained more than 50% of the variance in annual runoff ($r^2$ larger than 0.5).

For each catchment, we tested if the P-R relationship during the MD (April 2010 to March 2020) was different to the P-R relationship computed with the previous period (April 1979 to March 2010), by performing the analysis of variance model from R-package aov (Chambers et al., 2017) to the intercept parameter from the linear regressions (see Eq. 1 from Saft et al. 2015). From this analysis, we defined two types of cases: i) catchments with a significant shift in P-R relationship at a 0.1 significance level, and ii) catchments that did not experience a significant shift (test p-value greater than 0.1). For those catchments experiencing a significant shift, we computed the magnitude of the shift as the relative difference between

streamflow estimations from both linear regressions (prior the MD and during the MD), given the same precipitation value. This characteristic precipitation value was defined for each basin as the average precipitation during the MD period.

### 3.2.3 Propagation of extreme, severe, moderate and mild droughts

In addition to computing the shifts in P-R relationships (Sect. 3.2.2), which represent an overall catchment response during the MD period, we analysed annual drought propagation over the entire period of record (April 1979 to March 2020) based
on annual precipitation and streamflow anomalies (Sect. 3.2.1). Drought propagation during a hydrological year was represented by the contrast between streamflow and precipitation anomalies.

In particular, we focused on catchment responses during two types of events: i) single-year extreme and severe droughts, and ii) multiyear moderate and mild droughts. For each catchment, precipitation anomalies were classified by following the drought classification thresholds provided by McKee et al., (1995). In this way, annual anomalies between 0.5 and 0.067
quantiles were classified as mild to moderate droughts. Annual anomalies below the quantile 0.067 were classified as severe to extreme droughts. These thresholds are currently being used by the Public Works Ministry to declare water scarcity decrees (DGA resolution No 1.674 from 2012).

To analyse the hydrological memory effect on the propagation of extreme and severe droughts, we separated these events based on the precipitation anomaly of the preceding year (below or above the median).

## 4 Results

### 4.1 Hydrologic regimes and catchments characteristics

Based on the classification scheme described in Sect. 3.1.1, we identified 72 pluvial and 34 snow-dominated basins. Some of their main characteristics are presented in Fig. 2. Most of the snow-dominated basins are located in central Chile (30º-35ºS), given the higher elevation of the Andes at these latitudes (Fig. 2a). These basins feature mean annual precipitation values
between 258 and 1882 mm (mean of 779 mm), with a markedly concentration of precipitation during the fall and winter months (April to September, Fig. 2b), while most of the streamflow is released during spring and summer months, when snowmelt occurs (October to Jan, Fig. 2b). Pluvial basins, mostly located towards the south of the study region (35º-41ºS), have mean annual precipitation values ranging from 428 to 3376 mm (mean of 1862 mm), with a significant water accumulation (mostly rainfall) outside the fall-winter season of maximum precipitation. Streamflow in pluvial basins follows
the seasonality of precipitation closely (Fig. 2c).

To visualise the importance of groundwater processes within the study catchments, and their relationship with snow processes, in Fig. 2d we relate the GWI computed from HBV simulations with the SF derived from ERA5-L for each basin (Table 1). These variables come from independent datasets and show a significant correlation ($r^2 = 0.49$), which indicates the interdependence of snow and groundwater processes. The HBV model calibration resulted in overall acceptable simulation
performance in snow-dominated and pluvial basins, with NPE values of 0.72, 0.88 and 0.95 for the 10[th], 50[th] and 90[th]

percentile, respectively. For comparison, the corresponding Kling-Gupta efficiency values (Gupta et al., 2009) were, respectively, 0.57, 0.8 and 0.92.

In addition to the climatic characteristics (e.g., precipitation, snow fraction and aridity), GW contribution to runoff depends on physical factors (e.g., geology, topography, soil properties, soil drainage density), which may explain the large scatter in

Fig. 2d. In fact, the geologic characteristics vary across basins, as can be seen in Fig. S1. The most common geologic classes in snow-dominated basins are acid volcanic rocks (main class in 59% of basins), followed by acid plutonic rocks (main class in 18% of basins) and pyroclastics (main class in 18% of basins). In pluvial basins, there is greater heterogeneity in geologic classes, with 22% of basins dominated by pyroclastics and 19% of basins by acid plutonic.

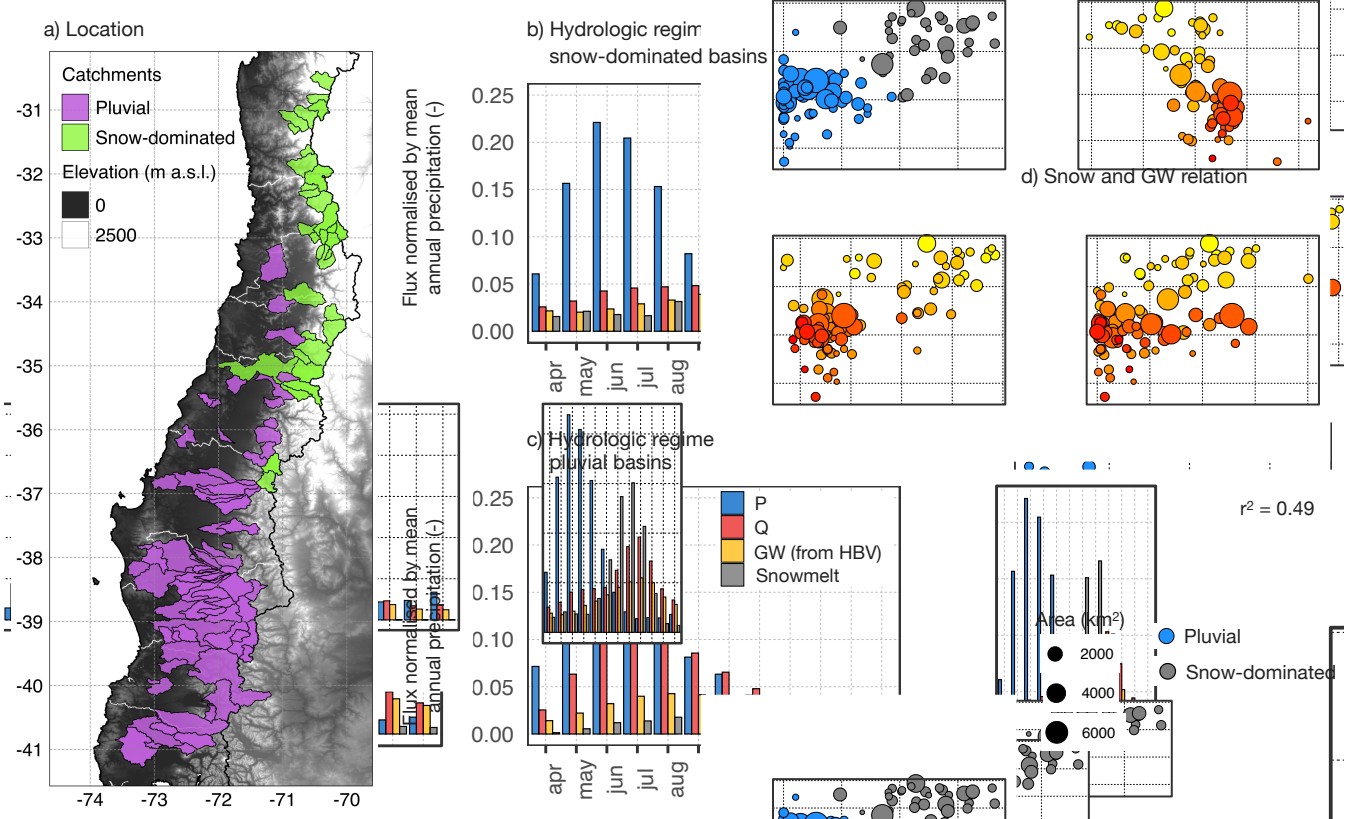

**Figure 2: Characteristics of pluvial and snow dominated basins obtained from CAMELS-CL dataset. Panel a presents the catchments location. Panel b and c present the hydrologic regimes for snow-dominated and pluvial catchments, respectively. The mean monthly fluxes of precipitation (P), streamflow (Q), groundwater (GW) were normalised by the annual P from CAMELS-CL dataset for the period April 1979 to March 2020. Since snowmelt was obtained from a different dataset, the mean monthly snowmelt fluxes were normalised by the mean annual precipitation from ERA5-Land. Panel d shows the relation between GWI**
**and SF, representing groundwater and snow storages, respectively.**

Additional characteristics of snow-dominated and pluvial catchments, including $P_{A-S}$ ratio, $Q_{A-S}$ ratio, SF, elevation, area, baseflow index, main land cover class, and granted water used rights are presented in Fig. S1. By construction, there is a clear distinction in $P_{A-S}$ ratio, $Q_{A-S}$ ratio, and SF from both clusters, which is translated into differences in mean catchment

elevations (mean elevation of snow-dominated catchments is 2667 m a.s.l., while mean elevation of pluvial basins is 599 m
a.s.l.). Both clusters present similar areas, and pluvial basins present lower baseflow indices compared to snow-dominated
basins. Higher baseflow indices in snow-dominated basins are consistent with the larger GWI simulated in these basins (Fig.
2d).

Regarding the land cover, 68% of the snow-dominated catchments are mainly covered by barren soil and snow, while the
rest is mainly covered by shrubland. None of these land cover classes is directly associated with anthropic activities. 94% of
the snow-dominated basins have less than 5% of their areas covered by croplands.

The region where pluvial basins are located features a higher heterogeneity of land cover classes, compared to Andean
region of central Chile (where snow-dominated basins are located) (Alvarez-Garreton et al., 2018). The dominant land cover
classes in pluvial basins are native forest (main class in 61% of pluvial basins) and shrubland (main class in 11% of pluvial
basins). Anthropic-related land cover classes dominate the rest of the pluvial catchments: forest plantation in 11% of basins,
grassland in 9% of basins, and cropland in 7% of basins.

## 4.2 Hydrological memory

The correlation between fall-winter precipitation ($P_{A-S}$) and the observed streamflow and simulated GW over the following
seasons are presented in Fig. 3. $P_{A-S}$ was used instead of annual precipitation since it is more directly related to snow
dynamics. Precipitation in fall and winter represents between 50 and 100% of the total precipitation volume (Garreaud et al.,
295    2017).

Figure 3a indicates that $P_{A-S}$ explains more than half of the variance in spring and summer streamflow ($Q_{OND}$ and $Q_{JFM}$,
respectively) in snow-dominated catchments, which is consistent with the seasonal-lag hydrological memory produced by
snow accumulation and melting processes. In general, the $P_{A-S}$ control on streamflow in these basins is stronger during spring
and decreases towards summer (larger blue bars compared to cyan bars in Fig. 3a). This memory effect drastically decreases
during the subsequent fall season ($Q_{AMJ}$, represented by the green bars in Fig. 3a) for most snow-dominated basins, which
indicates that streamflow during fall is more influenced by the current seasonal precipitation than by the precipitation during
the preceding year. These correlations, representing memory times of 3 to 9 months (Sect. 3.1.3), are consistent with the
streamflow and snowmelt seasonality presented in Fig 2b.

Interestingly, the correlation between $P_{A-S}$ and the spring and summer GW generally increases towards the summer months
(Fig. 3b). This indicates a longer response time of GW to the solid fall-winter precipitation, compared to the streamflow time
response. Consistent with this longer GW memory, the correlation to $P_{A-S}$ does not decrease so drastically after summer as
streamflow does (larger green bars in Fig. 3b compared to Fig. 3a). Since snowmelt is mostly finished by January, the high $r^2$
values (up to 0.75 in some catchments) indicate that fall-winter precipitation contributes to the following year's runoff as
after being stored as groundwater (Fig. 2b).


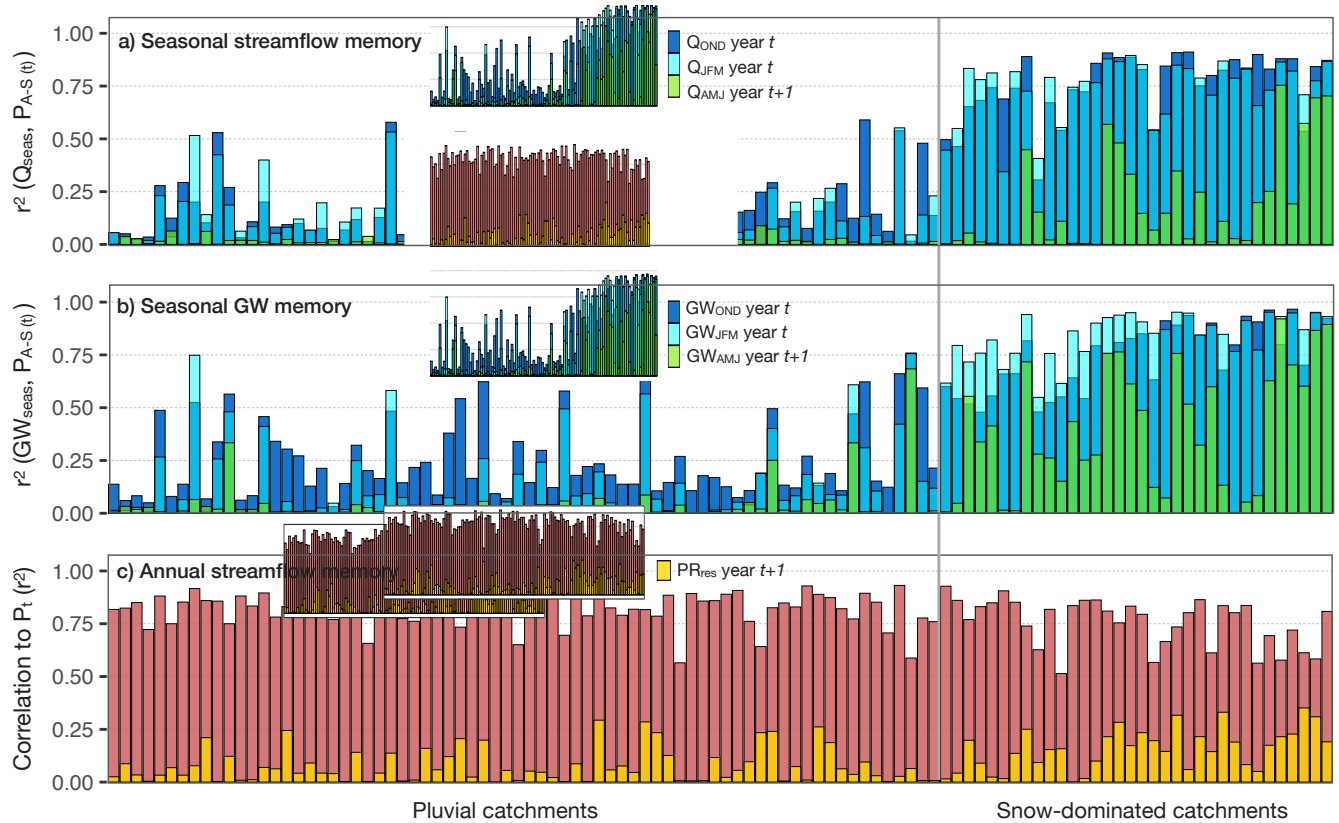

**Figure 3: Hydrological memory represented by different indices. Panel a presents the variance in seasonal streamflow explained by the variance in fall-winter precipitation of year t ($P_{A-S(t)}$). Panel b presents the seasonal GW correlation to $P_{A-S(t)}$. Panel c shows the $r^2$ between annual precipitation and annual streamflow ($r^2$ of P-R regressions), and the variance of the P-R residuals explained by the precipitation from the preceding year.**

The control of $P_{A-S}$ on subsequent seasonal streamflow is consistently weaker in pluvial catchments compared to snow-dominated ones, which is expected given the more direct runoff generation from rain. Also, in pluvial basins, the spring precipitation is larger than in northern basins and explains most of the streamflow generated during the same season, hence overshadowing the streamflow dependency to the previous season precipitation. The spring GW memory in most pluvial catchments is stronger than the streamflow memory (larger blue bars in Fig. 3b compared to 3a), which indicates a 3-month GW memory after winter precipitation. The GW memory during spring decreases towards summer (cyan bars in Fig. 3b) and mostly disappears towards the fall of the following year (green bars in Fig. 3b close to zero).

To remove the effect of the precipitation of the current year, Fig. 3c presents the dependency of the P-R regression residuals from Sect. 3.2.2 (i.e., annual streamflow not explained by the current precipitation) to the precipitation from the previous year. The $r^2$ values in Fig. 3c (all corresponding to positive correlation coefficients; not shown here) represents the control that the precipitation from the previous year exerts on the annual streamflow, and thus indicate hydrological memory beyond one year. Interestingly, although the residual $r^2$ are larger in snow-dominated catchments, there are also significant values in

the pluvial basins, suggesting that residence times associated to GW processes in pluvial catchments also contribute to more extended catchment memory.

The relationship between hydrological memory beyond one year and catchment storage dynamics driven by GW and snow processes is further explored in Fig. 4. The GWI derived from HBV simulations (Sect. 3.1.2) and SF (Table 1) are used to summarise GW and snow processes within a catchment in Fig. 4. These plots show that both snow and GW mechanisms contribute to the hydrological memory, which is consistent with the time lags these processes introduce to the pathways of precipitation within a basin. The adopted approach to compute the memory of a catchment, based on seasonal streamflow

and GW flows at the catchment outlet, represents the composite response times of all catchment mechanisms. Therefore, there might be other factors contributing to the overall response time of a catchment, including topography, soil properties, geology, drainage area and water table levels (Robinson and Ward, 2017), which may explain the large scatters in Fig. 4.

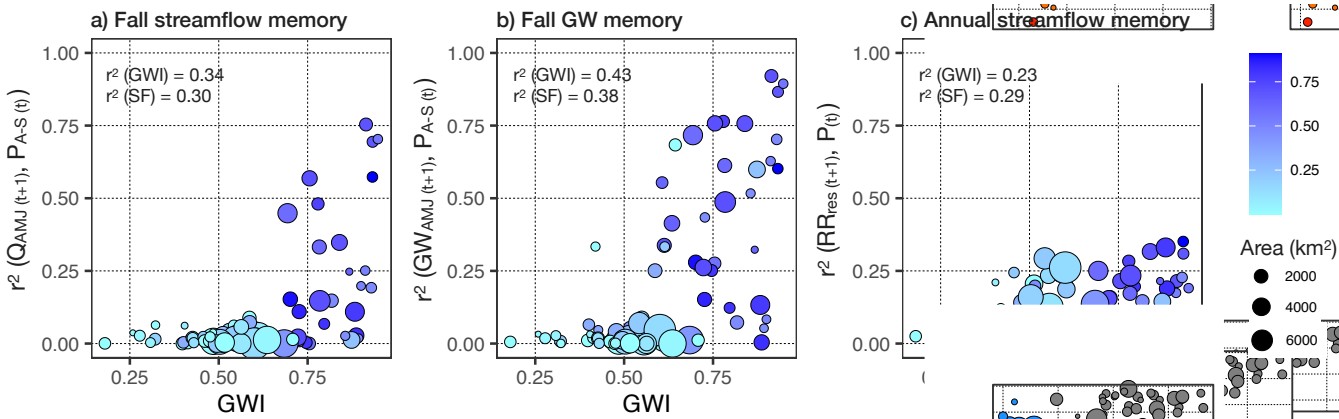

**Figure 4: Relationship between hydrological memory and indices of groundwater and snow storages (GWI and SF, respectively).**
**Panel a shows the fall streamflow (9-months) memory to the fall-winter precipitation of the preceding year. Panel b shows the fall GW (9-months) memory to the fall-winter precipitation of the preceding year. Panel c shows the (>12-months) memory of streamflow the precipitation of the preceding year.**

### 4.3 Droughts over the last decades

Given the markedly north to south precipitation gradient (Fig. 1), droughts characteristics in this section are analysed at the
catchment scale, but following a latitudinal order. In the following sections, we analyse drought propagation per hydrologic regimes. Heatmaps in Fig. 2 illustrate the precipitation and streamflow annual anomalies (Fig. 5a and 5) and z-scores (Fig. 5c and 5d). The unprecedented dry conditions during the last decade are evident. The temporal and spatial extent of precipitation deficits is shown by general negative anomalies for the 106 catchments between 30°S and 41°S (Fig. 5a). Over the previous three decades, there are few analogues showing such spatial pattern, such as the 1988-1990 three-year drought,
and the single-year droughts in 1996 and 1998 (one of the driest years in the last millennium, Garreaud et al., 2017) and 2007. The decade-long persistence of the MD spatial pattern, however, has no analogue in the studied period, nor in the last century (Garreaud et al., 2017).

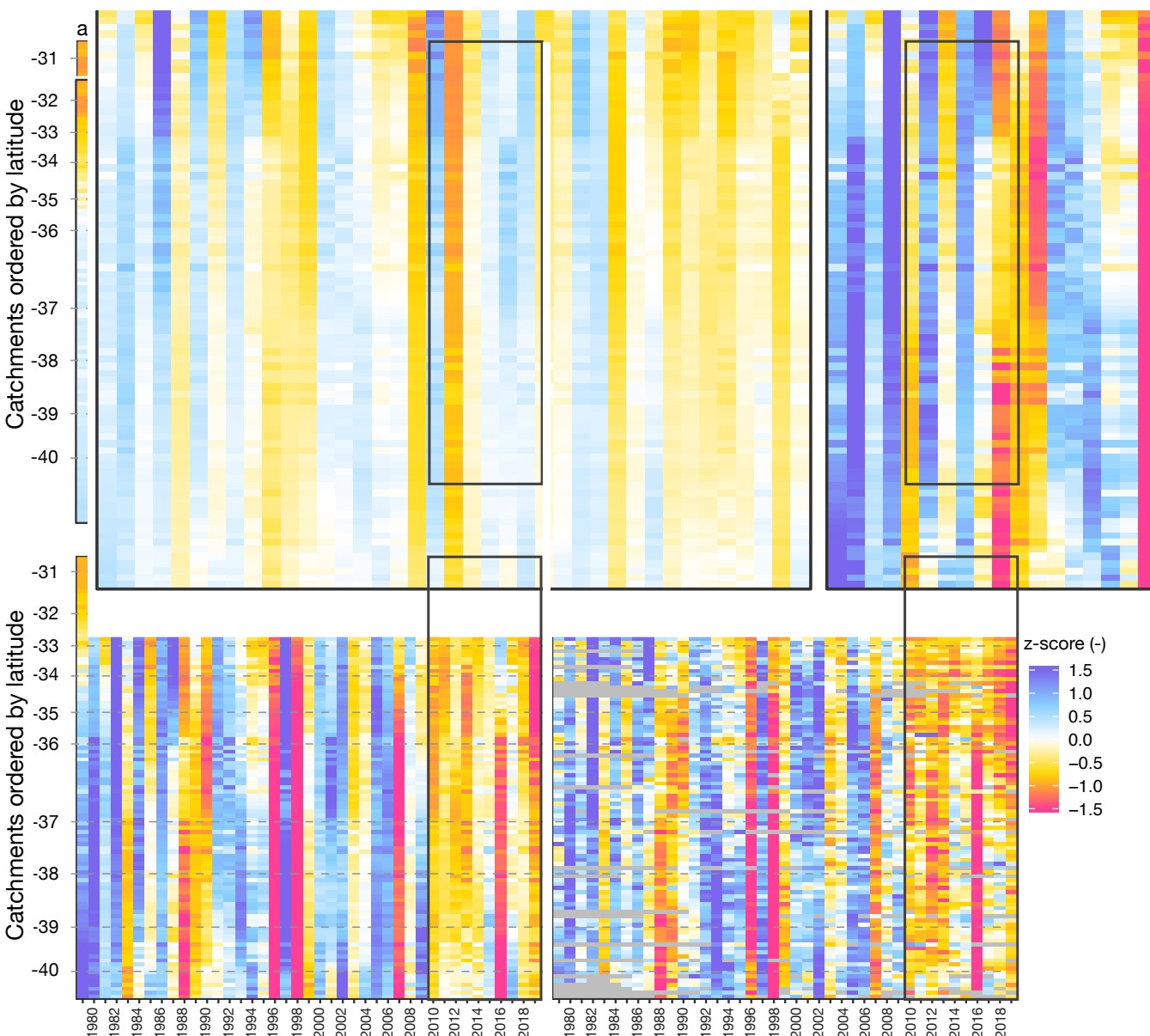

**Figure 5: Relative annual anomalies of catchment-scale precipitation (panel a) and streamflow (panel b). Panels c and d present the z-score of annual precipitation and streamflow. The MD period (April 2010 to March 2020) is highlighted in a grey box. Each row in the heatmaps corresponds to one study catchment and the catchments are sorted from north to south to illustrate regional drought patterns.**

The relative precipitation anomalies are consistently higher in catchments north of 35ºS (Fig. 5a), which is partly due to the very low annual precipitation values in the northern region compared to the southern region. The probability of having high anomalies is larger in this northern region (z-scores closer to zero in Fig. 5c), where few meteorological events contribute to most of the annual accumulation. Thus, the interannual variability is higher than in the southern region (south of 35ºS, where

most pluvial basins are located, Fig. 2), where several precipitation events occur during the year. During the MD, annual precipitation in basins south of 35°S decreased up to 1.5 times below the historical standard deviation of annual precipitation, which represents an exceedance probability of 94.3%. Z-scores during the MD have been usually lower in the southern region, compared to central Chile (30-35°S).

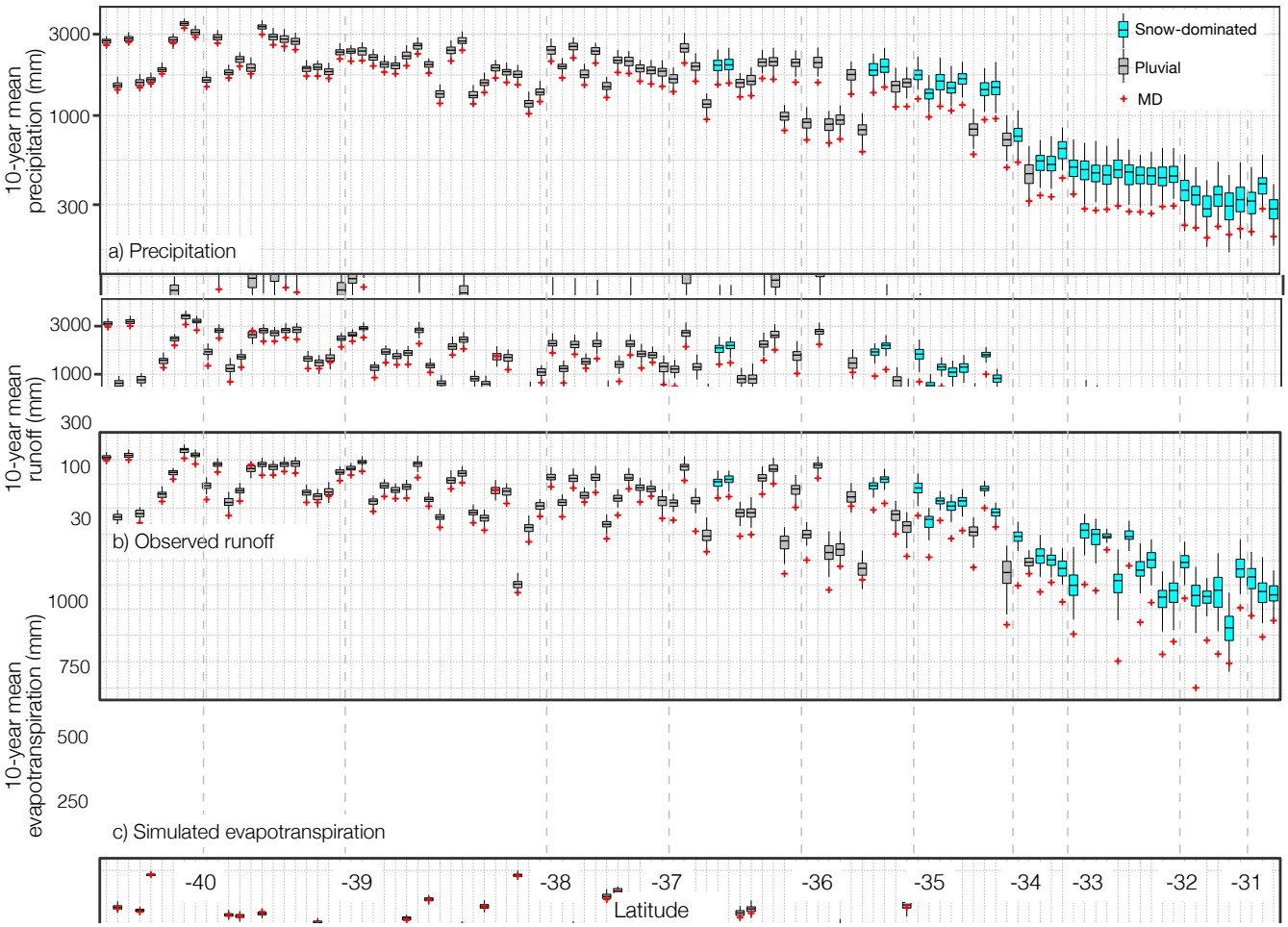

**Figure 6: Boxplots of 10-year mean precipitation (panel a), observed runoff (panel b) and simulated ET from HBV (panel c) for the period April 1970 to March 2010 (i.e., excluding the MD). The red marks correspond to the decadal means during the MD (April 2010 to March 2020). Please note the log-scale of the Y-axes in panels a and b.**

The annual streamflow relative anomalies (Fig. 5b) present larger values and larger variations in space and time than precipitation, which is due to the lower absolute values compared to precipitation and the dependency of streamflow on local terrestrial characteristics. If we compare the spatial patterns of anomalies (Fig. 5b) and z-scores (Fig. 5d) during the MD, we see that consistently with precipitation, larger anomalies have been observed in catchments north of 35°S (extreme dry years up to 90% of streamflow deficits with a median of 57%). Still, the moderate deficits experienced in the southern region

(median deficit of 25%) have a lower probability of occurring (lower z-score values, indicating higher exceedance probabilities).

To further characterise the hydro-climatic anomalies during the MD, Fig. 6 presents the frequency distribution of 10-year mean precipitation, observed streamflow and simulated ET for each basin, with the mean values during the MD plotted in red dots. These plots indicate that the MD has been extremely unusual in terms of precipitation and streamflow. The average

precipitation during the MD is within the first decile for 91% of the study catchments (96 out of 106). The average runoff during the MD has been more extreme than precipitation deficits, with 96% of catchments (102 out of 106) presenting 10-year mean runoff values within the first decile. These values represent the minimum value over the last four decades for some basins located north to 32ºS and correspond to an outlier of the historical distribution for the rest of the catchments. The average 10-year ET during the MD has been less unusual than precipitation and streamflow anomalies, with only 20%

of catchments (21 out of 106) within the first decile, and 73% of basins below the median (quantile 0.5) historical 10-year ET values. This indicates that in general, the estimated ET within the study basins during the MD has not been as different from normal conditions as precipitation and streamflow. This also shows that, despite the higher mean temperatures due to the warming trend experienced in the region (Boisier et al., 2018), ET is closely related to precipitation, notably in the northern (water-limited) basins.

**4.4 Shifts in P-R relationships during the MD**

Given the relatively small scale of catchments in Chile, runoff in most of them has a strong dependency to the interannual precipitation variability, explaining typically ~ 75% of the streamflow variances. Yet, the annual P-R relationships observed during the MD changed significantly for some of the catchments within the study region (examples in Fig. 7b, 7c, 7e, 7f), while other catchments showed no significant change (Fig. 7a, 7d). Table 2 summarises the number of catchments that

experienced a significant shift in P-R relationship during the MD, and their associated shift magnitudes. This table also shows the results for those catchments with no change. From the 106 studied catchments, 61.3% had a significant change in the P-R relationship during the MD, and 38.7% had not changed (Table 2).

For the 65 catchments showing a change, the historical P-R regressions consistently underestimate the runoff deficits during the MD. 95% (62 out of 65) of the catchments with a significant change in P-R relationship during the MD, had a negative

shift; that is, an intensification in drought propagation. For similar precipitation deficits as other dry years, observed streamflow during the MD in snow-dominated catchments were up to 57% lower than those predicted by the historical P-R relationship. In pluvial catchments, these shifts reached up to 37% (Table 1).

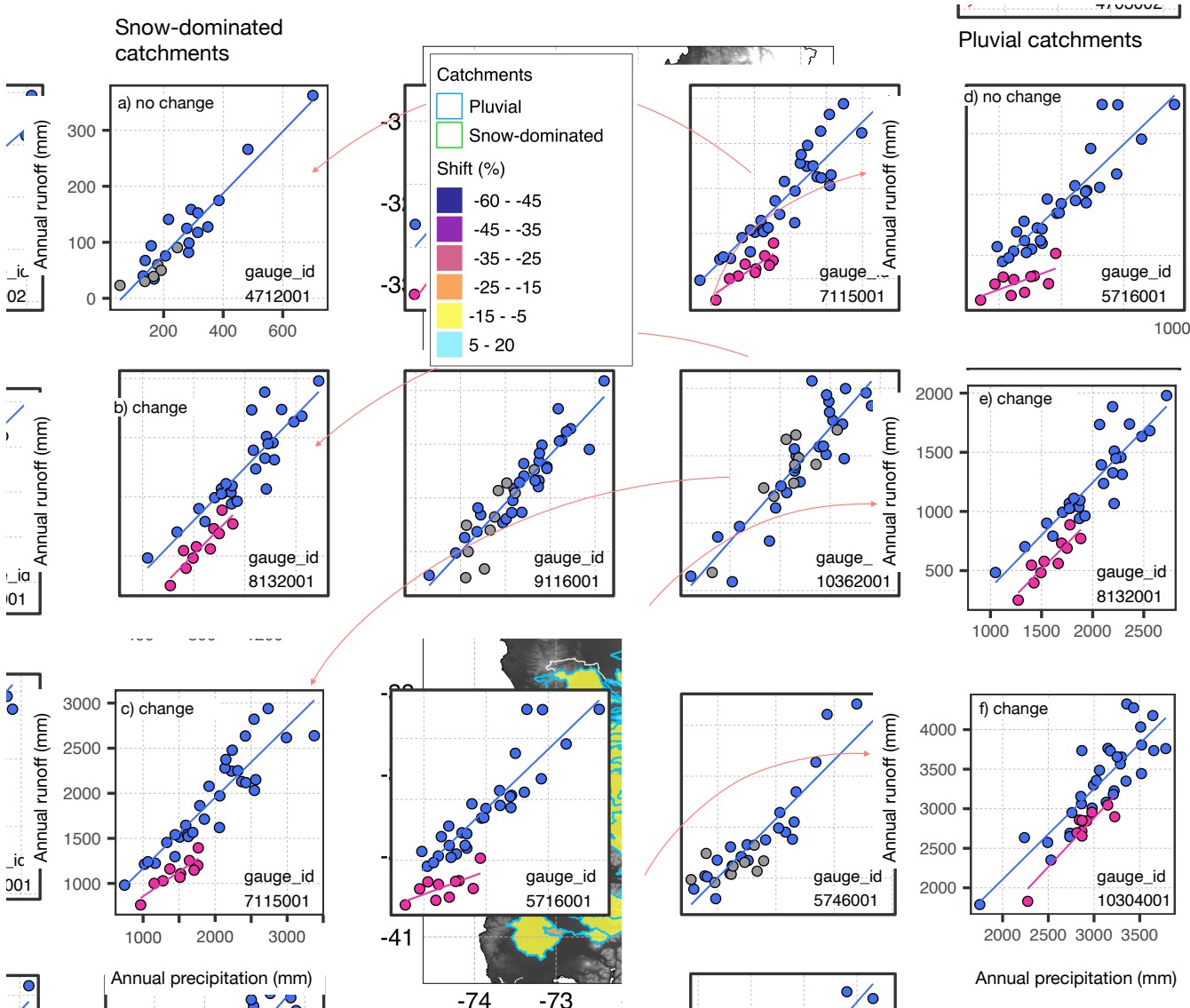

**Figure 7: Annual runoff (y-axes) and precipitation (x-axes) for selected snow-dominated catchments with and without a significant change in P-R relationship during the MD (panels a-c). Panels d-f show three selected pluvial basins with and without change. The years within the MD are highlighted in magenta when there was change and grey when there was no change in P-R regressions. The map shows the catchments with change coloured by their shift in P-R relationship.**

For those catchments with significant change, higher GWI values are associated to larger shifts in P-R relationships ($r^2$ = 0.40, not shown here), and thus to the intensification of drought propagation during the MD, with respect to historical annual responses to droughts. The relationship between snow fraction and P-R shifts is weaker ($r^2$ = 0.29, not shown here), probably because snow processes control the hydrologic response of a subset of catchments experiencing significant change in P-R relationship, whereas GW is inherent to all basins.

While these results provide insights about changes in P-R relationship during a multiyear period, they do not indicate if the changes are progressive, i.e., if the basins progressively generate less water for a given precipitation amount, compared with their historical behaviour. We address this in the following section, where annual drought propagation is analysed in detail.

**Table 2: Number of catchments with significant shift and no shift in P-R relationship during the MD. The shift magnitude in catchments with change is given in italics.**

|  | Snow-dominated | Pluvial | Total |
|---|---|---|---|
| Change<br>Mean shift (min, max) | 26<br>*-31% (-57%, -12%)* | 39<br>*-11% (-37%, 27%)* | 65<br>*-19% (-57%, 27%)* |
| No change | 8 | 33 | 41 |
| Total | 34 | 72 | 106 |

## 4.5 Annual drought propagation

Figure 8 presents the time series of annual precipitation and streamflow anomalies averaged across snow-dominated catchments (Fig. 8a) and pluvial catchments (Fig. 8b) over the last four decades. Drought propagation is represented by the difference between average runoff and precipitation anomalies (secondary y-axis in Fig. 8). To focus on drought propagation and not on the propagation of positive anomalies, the difference in runoff to precipitation anomalies was computed only for those years with negative streamflow anomalies (red line in Fig. 8).

In snow-dominated basins, the difference between runoff and precipitation anomalies consistently increases (becomes more negative) in the second year of precipitation deficits, showing that in catchments with longer hydrological memory consecutive years with precipitation deficits are associated with intensified drought propagation. This plot also provides insights about hydrological recovery, understood as the hydrological condition after a meteorological drought has ceased (Yang et al., 2017). While 2016 had near average precipitation in snow-dominated basins, probably there was not enough water entering the system over enough time to recharge groundwater systems up to levels such as those before the MD (similarly to the conceptual drought propagation illustrated in Fig. 3 from Van Loon, 2015). This is reflected by the larger streamflow deficits in 2016 compared to 2008, even when the above-mean precipitations in 2008 following the deficits in 2007, are comparable to those in 2016 and 2015, respectively. This can be related to the catchments' memory (Sect. 4.2) and the 7-year (2009 to 2015) precipitation deficits prior 2016, which probably prevented a full hydrological recovery after a single year of above-average precipitation. These results are consistent with large recovery times reported for semi-arid Australian catchments following extreme droughts (Fowler et al., 2020; Yang et al., 2017). In this way, hydrological memory would be an explanatory factor for both the intensification in drought propagation as well as a delayed hydrological recovery.

For pluvial catchments (Fig. 8b), prior to 2010, the propagation of meteorological to hydrological drought has ranged around 0 to 15% (i.e., streamflow anomalies have been, on average, up to 15% lower than precipitation anomalies), independently of the precipitation deficits of previous years. Such propagation was observed even in the driest two years of the historical record, 1996 and 1998. After 2010, there are two years (2012 and 2016) where drought propagation has been intensified up to 25%. These larger streamflow deficits during the MD may be due to different factors, including the large ET in 2012 and 2016 (positive anomalies and z-scores above 1.5 as can be seen in Fig. S2) combined with large precipitation deficits.

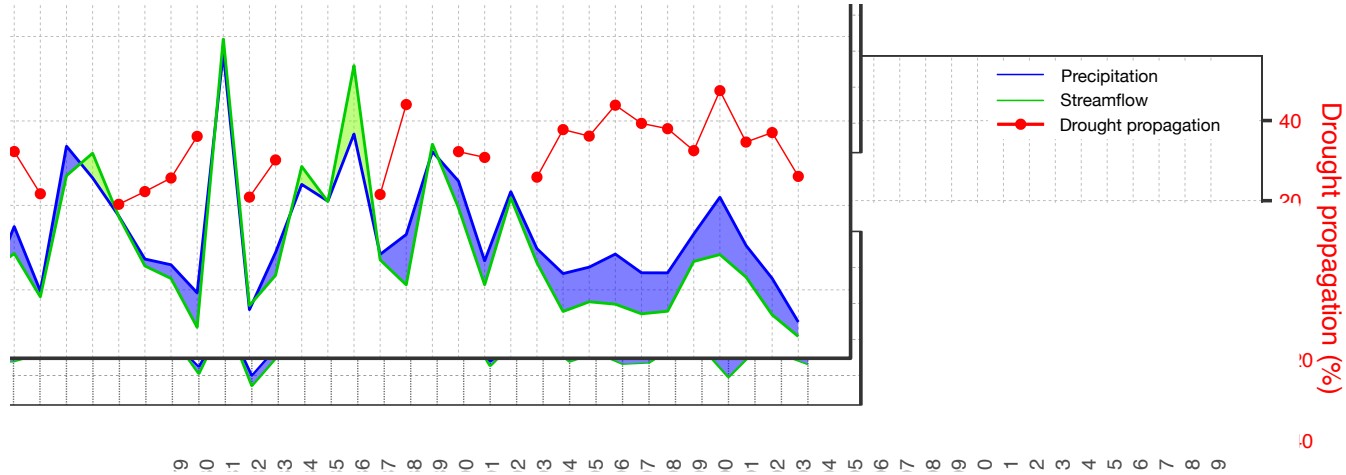

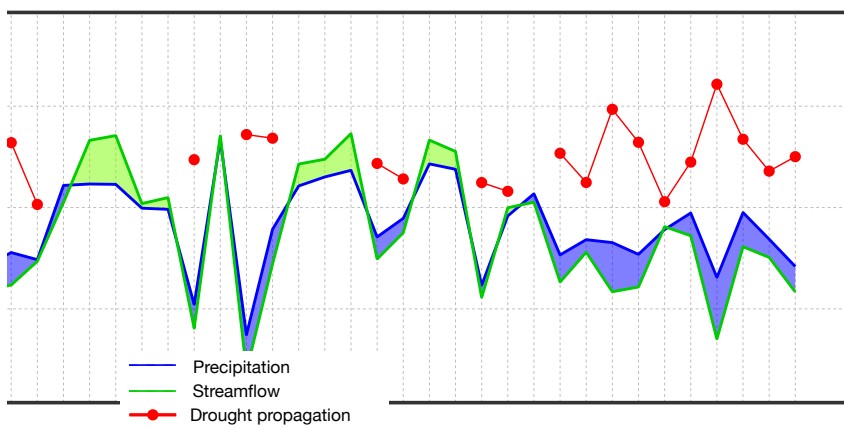

**Figure 8: Precipitation and runoff anomalies in snow-dominated (panel a) and pluvial catchments (panel b). The blue shaded area between curves represents precipitation anomalies larger than streamflow anomalies, and green shaded area represents the opposite. The meteorological to hydrological drought propagation, represented by the difference between runoff and precipitation anomalies for those years where runoff anomaly is below zero, is shown in the secondary y-axis by red points connected with red lines between consecutive years.**

Regarding the overall response during the MD, since P-R relationships do not explicitly account for variations in ET, the positive ET anomalies during specific years of the MD (2012, 2016 and 2018, Fig. S2) may partly explain the P-R shifts identified in some pluvial catchments (Fig. 7). Towards the south of the study region, basins move from water-limited to energy-limited (Alvarez-Garreton et al., 2018). Therefore, ET in pluvial catchments is modulated by both the available water and the available energy, in contrast to snow-dominated catchments, where ET is primarily driven by precipitation (these are water-limited basins). This suggests that ET may be a factor influencing the intensification of drought propagation during the MD in pluvial catchments. However, despite higher ET in three years during the MD, the average ET during the 10-year MD period has been lower than other 10-year windows (Fig. 6c). Therefore, the hydrological memory beyond one hydrological year in pluvial basins (Fig. 3) is likely another factor contributing to the P-R shifts in pluvial catchments.

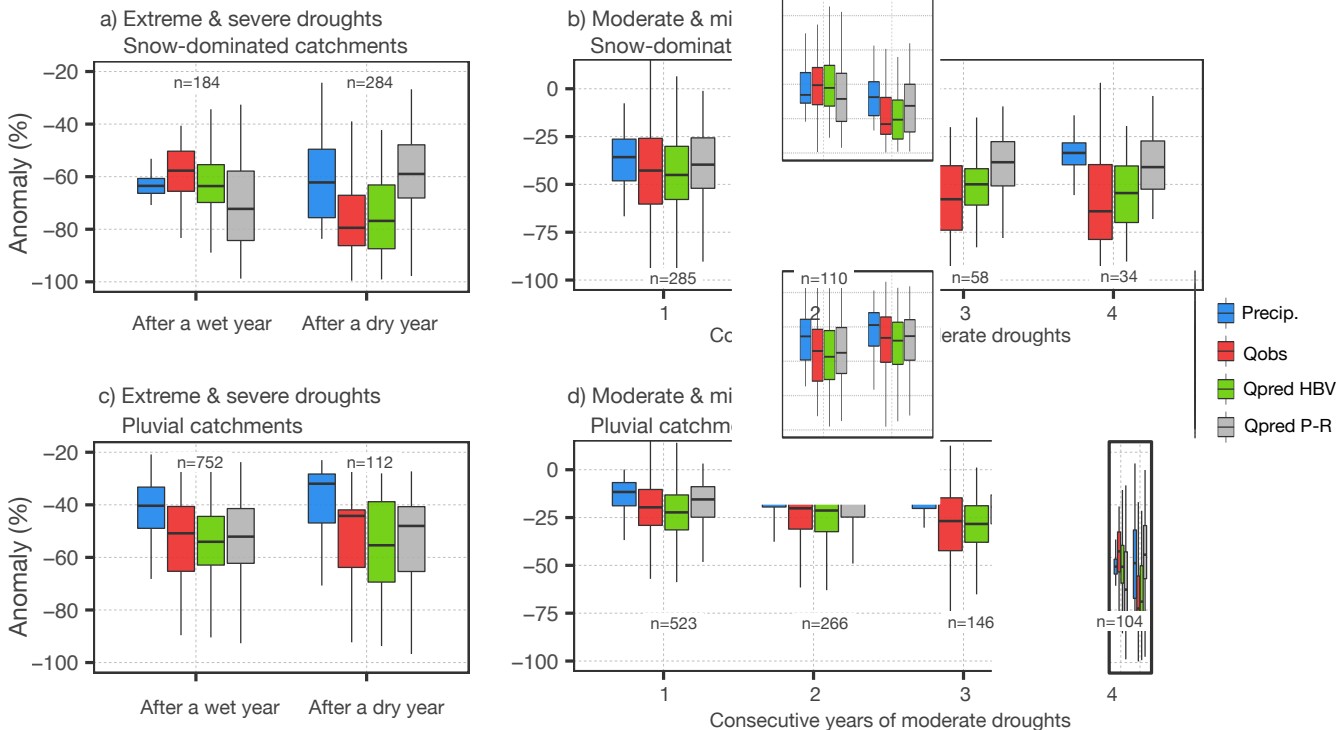

**Figure 9: Observed and simulated annual drought propagation for consecutive dry years. Panel a presents the results for snow-dominated catchments, and panel b for pluvial catchments. The number of cases is presented for each set of boxplots.**

Figure 9 presents the observed and simulated drought propagation during single-year severe and extreme droughts and during persistent mild and moderate droughts, for snow-dominated (Fig. 9a, 9b) and pluvial catchments (Fig. 9c, 9d). Consistently with the hydrological memory in snow-dominated catchments, we observe that drought propagation in these basins is highly dependent on the meteorological conditions from the previous year (Fig. 9a), which define the initial condition of soil water storages. If a severe or extreme drought happens after a wet year, such as 1981, 1985, 1988, 1998, 2007 and 2009 (Fig. 9a), drought propagates without amplification, i.e., streamflow deficits are lower than precipitation deficits. By contrast, if the extreme drought happens after a dry year, such as 1995, 1996, 2013 and 2019, meteorological

droughts are amplified by nearly 20% (difference between median streamflow and median precipitation deficits in Fig. 9a). This is also observed in persistent but moderate droughts, when under similar precipitation deficits, the surface water supply decreases after one year with below-average conditions.

If we look at streamflow predictions, these plots indicate that the HBV model represents catchment response for extreme and persistent droughts well, consistently outperforming the prediction from P-R regressions. It should be noted though that HBV allows memory effects only to a certain extent given the groundwater storage capacity defined by calibration.

Pluvial basins in the study region have shorter hydrological memory compared to snow-dominated catchments, which leads to a more similar behaviour under extreme meteorological droughts occurring after a wet and a dry year (difference in streamflow to precipitation anomalies between 10-20%, Fig. 9c). However, even when these basins are largely controlled by precipitation during the same year, there is some over-one-year memory (Fig. 3c), which may be influencing the observed decrease in streamflow generation after two years of consecutive precipitation deficits (Fig. 9d). This effect is well captured by the HBV model, while the annual P-R relationship tends to overestimate observed runoff. This indicates that a good representation of fluxes such as ET, soil moisture and groundwater dynamics, allows foreseeing catchment response to persistent droughts and to extreme droughts.

## 5 Discussion

Hydrological memory is related to slow groundwater and subsurface flows transferring precipitation from previous seasons, especially from winter (Fig. 3 and Fig. 4). If the winter snow pack is large, hydrological memory is further extended, not only by the season-lag resulting from the snowmelt contribution to streamflow in spring and summer, but also by the slow GW contribution during the next fall, when the snow has already melted.

We showed that the importance of GW contribution to runoff is positively correlated with snow accumulation (Fig. 2d). This relationship between snow and groundwater contribution to downstream streamflow supports the GW recharge conceptualisation recently proposed by Taucare et al. (2020) for the Western Andean Front in central Chile. In contrast to former assumptions of no GW recharge in high elevated Andean areas, Taucare et al. (2020) demonstrated the existence of GW circulation in fractured rocks originating from rain and snowmelt above ~2000 m a.s.l. (the mean elevation of the snow-dominated basins in this study range from 737 to 3783, with a mean of 2667 m a.s.l.).

This conceptualisation is also supported by the groundwater recharge mechanisms driven by snowmelt shown by Carroll et al., (2019) in a Colorado River headwater basin. Snowmelt infiltrates in situ, where ET is low compared to precipitation, the soil storage is shallow, and there is a low permeability bedrock underneath. Infiltrated snowmelt is routed through the steep topography as shallow ephemeral interflow, which supports large recharge rates in topographic convergence zones, where ET is still moderate and ephemeral stream channels in the upper basin appear (Anderson and Burt, 1978). At high elevations where snowmelt starts, topography and soil permeability would have a larger control in groundwater recharge than precipitation. That is, the rates at which snowmelt infiltrates are more sensitive to catchment characteristics than to the

snowpack volume (Carroll et al., 2019). This suggests that during dry years, the portion of snowmelt directly contributing to runoff would decrease while the portion of GW from snowmelt infiltration would increase.

The large snowmelt infiltration rates at high elevations during snowmelt season, combined with the absence of precipitation events (precipitation is concentrated in winter season) would explain the higher GWI in snow-dominated basins.

High snow fraction ratios also indicate an important spatial variation of precipitation within the basin, with most precipitation falling in the upper part of the basin and therefore, traveling longer paths to reach the outlet where streamflow is recorded. Longer paths lead to longer travel times (i.e., longer hydrological memory), which may support groundwater

recharge from interflow.

Regarding drought propagation, our results show that precipitation influence for longer times in basins where snow and GW processes dominate the hydrological response. Snow-dominated catchments feature longer memory than pluvial basins, and have been more affected by the persistency of precipitation deficits during the MD, causing a significant change in overall catchment response to the precipitation over the last decade. These results complement previous findings in Australia (Saft et

al., 2015, 2016b, 2016a), providing new evidence of the vulnerability of catchments to drying climatic trends. In particular, our results reveal that snow processes are tightly associated to the intensification in drought propagation, an insight that was not drawn by the analysis of the Australian Millennium Drought given the lower elevation of the analysed Australian basins (mean elevations below 1500 m a.s.l.).

The role of snow in hydrological memory and its impacts on water provision during droughts highlights the need to further

understand the seasonal characteristics of drought propagation. The effects of spring-summer precipitation deficits may be different from those of fall-winter winter deficits (Berghuijs et al., 2014; Jasechko et al., 2014). This is particularly important in central-south Chile, where shifts in precipitation and streamflow seasonality due to anthropogenic climate change have already been detected in observations (Boisier et al., 2018; Bozkurt et al., 2018; Cortés et al., 2011).

Our results also provide insights regarding model representation of hydrologic mechanisms. The HBV model overcome

some of the limitations of diagnosing the progressive water deficits only from shifts in annual P-R relationships. A model that can capture catchment memory is more suited to represent deficits in streamflow under persistent drought, compared to the simulations from P-R regressions that only consider the precipitation of the same year (Fowler et al., 2020). Nevertheless, the HBV model is a simplified representation of actual hydrological processes, which limits its capability to simulate long-term memory effects. Memory effects in the HBV model are caused by soil water storage as this store can

accumulate precipitation deficits. The groundwater stores in HBV cannot drain below the level where streamflow ceases and, thus, represent only the dynamic storage in a catchment (Staudinger et al., 2017), which means that memory effects caused by groundwater stores are limited in the model. This is consistent with the limitations of conceptual bucket-type models to simulate long, slow hydrological processes demonstrated by Fowler et al., (2020). The effects of multiyear drought cannot accumulate in models where the time constants (usually months) are shorter than the observed memory (seasonal and

longer). In this way, catchments that are more prone to a non-stationary hydrologic response under persistent droughts, pose

greater challenges for projecting the impacts of a drying climate. This highlights the need to advance towards robust modelling frameworks in order to achieve reliable streamflow predictions under drier climate projections.

The proposed approach focuses on understanding the causes of intensified drought propagation by analysing the basins runoff mechanisms and the hydrological memory within the basins. However, anthropic factors such as irrigated agriculture,

reservoirs and water abstractions for human consumption may also contribute to different catchment responses to precipitation during persistent droughts. While the water used by natural ecosystems are related to the incoming precipitation (e.g., in water-limited basins ET is proportional to the precipitation), the water used to supply human demands may remain similar to pre-drought conditions if the associated infrastructure allows it (e.g., deep pumping wells or large reservoirs). This may cause a decrease in runoff which is not directly related to precipitation deficits. In this study, we aimed at removing

some of these anthropic effects by filtering out catchments with reservoirs, but there are still anthropic-activities within the analysed sample. In particular, pluvial basins feature human-induced land cover classes such as cropland, forest plantations and grassland (Sect. 4.1). Previous findings have related these classes to differences in water provision, particularly during dry years (Alvarez-Garreton et al., 2019). Hence, drought propagation within these basins is likely a combined result of climatic conditions, hydrological mechanisms and anthropic effects. In contrast, the sample of snow-dominated catchments

are less prone to anthropic effects given their location in high elevated areas of the Andes cordillera (no human-related dominant land cover classes within the basins, Sect. 4.1). This suggests that the intensification of drought propagation in these basins is mainly due to climatic conditions and hydrological mechanisms.

## 6 Conclusions

Our analysis of 106 basins along central Chile indicates that larger solid precipitation fraction within a catchment leads to

increased slow GW contribution to runoff thus connecting precipitation anomalies in a given winter with streamflow until fall of the next year. In this way, snow-dominated catchments have a memory that strongly connects streamflow generation over consecutive hydrological years. In pluvial basins on the other hand, hydrological memory is shorter and the annual streamflow is mostly explained by the precipitation of the current year.

These different hydrological memories have led to contrasted drought propagation during the MD. The MD in central-south

Chile has been extraordinary because of its persistence (10 years to date) and extended spatial domain (~1000 km). Annual precipitation anomalies during the MD have been larger in snow-dominated catchments (mostly located between 30ºS-35ºS) compared to pluvial catchments (mostly located between 35ºS-41ºS). However, observing such large annual deficits in the northern region is not unusual given the highly variable rainfall regime. The annual precipitation anomalies experienced in the southern region, on the other hand, have been highly unusual with respect to the last four decades.

Catchments with longer hydrological memory showed larger shifts in P-R relationships during the MD, compared to their historical behaviour, revealing the intensification of drought propagation during multiyear droughts. For snow-dominated catchments, after one year of precipitation deficits, the surface water supply –under equivalent precipitation deficits–

significantly decreases. That is, the basins progressively generate less streamflow for a given precipitation amount compared with the historical behaviour, because snowmelt infiltrates into depleted levels of GW and does not reach the catchment outlets during the same hydrological year. In pluvial basins on the other hand, there is a weaker decrease in the water supply after consecutive years of precipitation deficits.

What is worse, an extreme single year drought or a persistent moderate drought? We have shown that for any type of drought, hydrological memory and initial storages conditions are key factors modulating catchment responses. In snow-dominated basins located in the Andean semi-arid region of Chile, catchments strongly depend on both the current and previous precipitation seasons. In absolute terms, single year extreme droughts induce larger absolute streamflow deficits (i.e., less water supply). However, moderate but persistent deficits induce a more intensified propagation of the meteorological drought (larger streamflow deficits relative to precipitation). The worst scenario would be an extreme meteorological drought following consecutive years of below-average precipitation, as occurred in 2019. In pluvial regimes, initial conditions and hydrologic memory are still important factors to represent catchment response fully. However, water supply is more strongly dependent on the meteorological conditions of the current year. Therefore, an extreme drought would have a higher impact on water supply than a persistent but moderate drought.

Snow-dominated catchments store water that is then released during dry seasons, a characteristic particularly valuable in regions with a limited water supply and a severe risk to droughts, such as in central Chile. We have demonstrated that these basins are prone to intensify the propagation of persistent droughts, which pose additional challenges to water management adaptation in central Chile given the drying projected trends for the region.

*Data availability*. CAMELS-CL catchment dataset was obtained from the Center for Climate and Resilience Research website (http://www.cr2.cl/camels-cl/). ECMWF ERA5-Land dataset was downloaded from https://www.ecmwf.int/en/era5-land. ASTER GDEM elevation data were downloaded from the NASA Earthdata website (https://search.earthdata.nasa.gov/search/).

*Author contributions*. This research was conceived and design by CAG, JPB and RG. JS and MV implemented the HBV model. CAG wrote the manuscript with input from all co-authors. All the authors have been involved in interpreting the results, discussing the findings, and editing the paper.

*Competing interests*. The authors declare that they have no conflict of interest.

*Acknowledgements*. This research has been developed within the framework of Center for Climate and Resilience Research (CR2, ANID/FONDAP/15110009) and the joint research project ANID/NSFC190018. CAG also acknowledges for support by ANID/FONDECYT/1201714. We thank the editor Markus Hrachowitz and the referees Anne Van Loon and Gemma Coxon for their constructive comments that greatly improved the original paper.

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

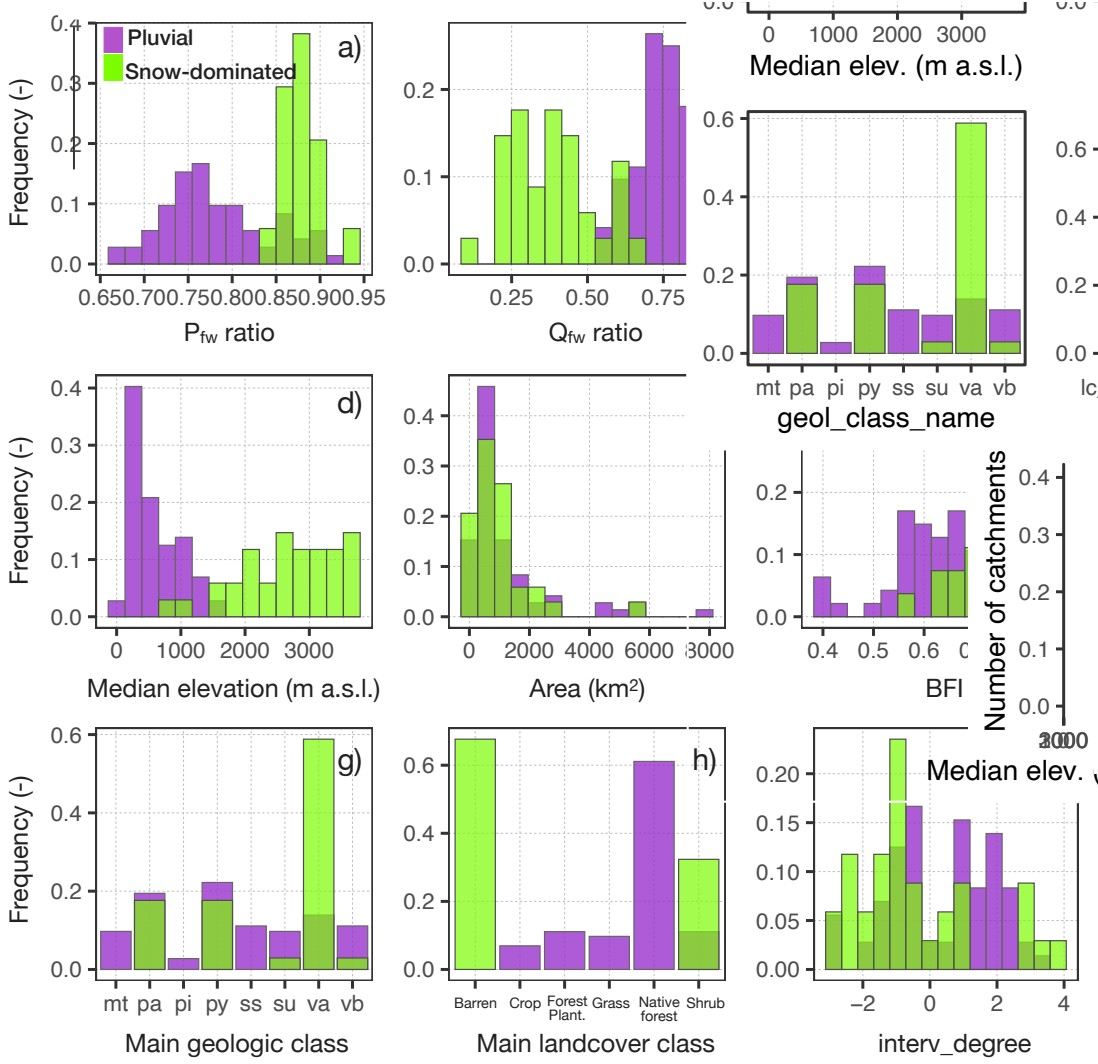

**Figure S1: Main characteristics of snow-dominated and pluvial catchments, including P$_{A-S}$ ratio (panel a), Q$_{A-S}$ ratio (panel b), SF (panel c), mean elevation (panel d), area (panel e), baseflow index (panel f), main geologic class (panel g), main land cover class (panel h), and granted water used rights (panel i).**

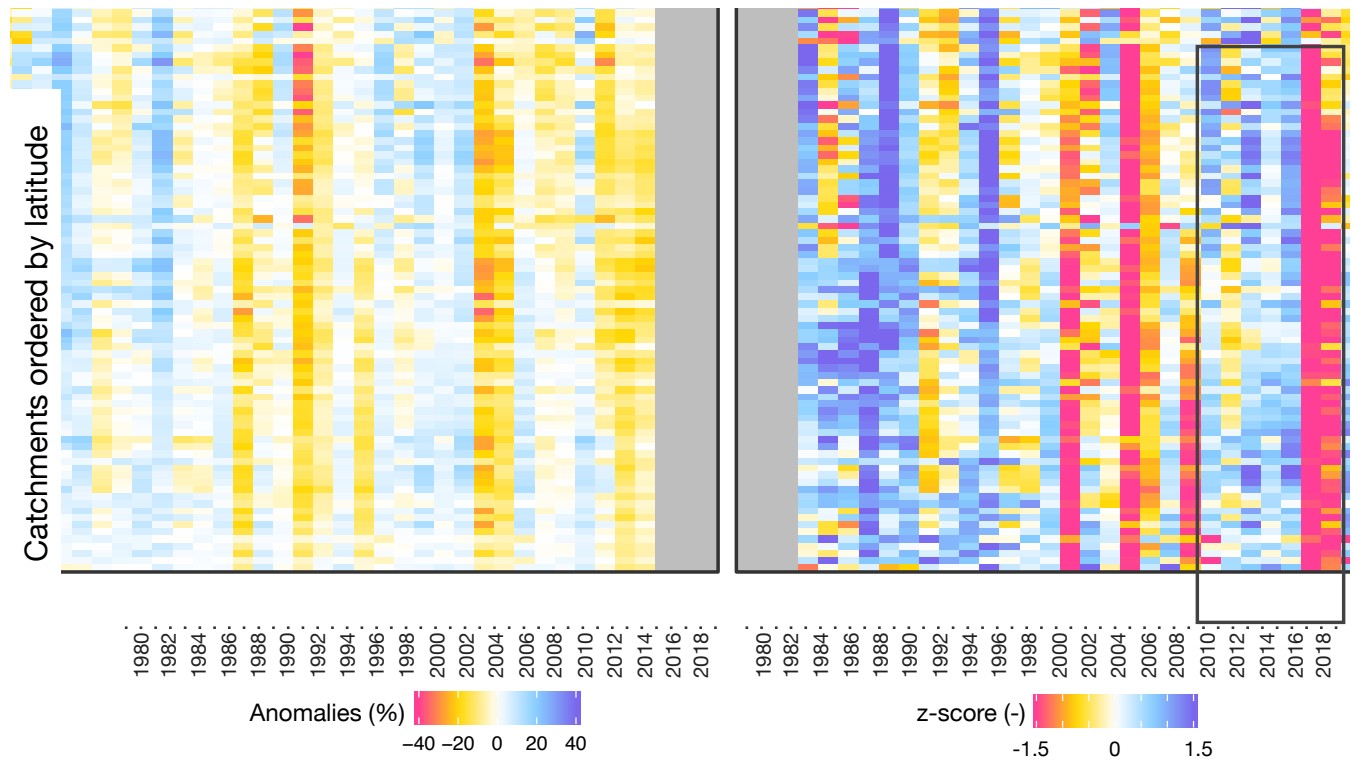


**Figure S2: Panel b presents the z-score of simulated annual ET (computed as deviations from mean normalised by standard deviation). The MD period (April 2010 to March 2020) is highlighted in a grey box. Each row in the heatmaps corresponds to one study catchment and the catchments are sorted from north to south to illustrate regional patterns.**