# Peer review of "Progressive water deficits during multiyear droughts in basins with long hydrological memory in Chile."

_Hydrology and Earth System Sciences, 2020_

## Referee Comment (RC1) · Anne Van Loon (Referee) · 5 Aug 2020

**Review of Camila Alvarez-Garreton et al. "Progressive water deficits during multi-year droughts in central-south Chile"**

- by Anne Van Loon

This paper uses a recently published dataset to look at changes in rainfall-runoff relationships during multi-year drought covering a range of different conditions in Chile. The authors conclude that both groundwater and snow play an important role and that some regions are more affected by prolonged drought and others by short extreme drought. The topic of the paper is important and the range of catchments included in the dataset provides interesting insights. The analysis is done well and the paper is generally well-written. The paper can be suitable for publication in HESS after a number of (relatively minor) revisions. Below I provide my suggestions for improvement of the paper.

**General comments**

The authors do not mention potential human influence in the catchments (except in the final sentence of the manuscript, l.461). It is unclear whether these catchments are completely without any human influence on the hydrology (which I doubt because they are quite large and cover a large part of Chile, so they are unlikely not to include reservoirs, forestry plantations, agriculture). There might for example be an increase of abstraction or change of land use during multi-year drought that might influence the rainfall-runoff relationship. CAMELS-CL includes information on land use, intervention degree, water rights and it would be greatly improve the paper if this was included in the analysis. If this cannot be included in the current paper, the authors must at least mention the degree of human influence in the catchments at the start of the paper and discuss potential effects of human influences on the results at the end (in the Discussion section).

The authors should be more specific about the role of snow vs. groundwater when they talk about memory (for example in the abstract l.16-18). This starts with the classification of catchments in semi-arid vs. temperate catchments, which seems to be partly overlapping with snow-dominated vs. rainfall-dominated catchments but this is not clearly identified. For example, on p.5 l.111-113 the authors state that in central Chile there is an Mediterranean regime, whereas in snow-dominated basins "streamflow peaks in spring / early summer", which makes me think that the snow-dominated basins are located in southern Chile. But then the classification (l.113-118) seems to be only based on precipitation and not on snow and Fig.1 shows that there is more snow in the north. Also, an additional classification of catchments is introduced on p.7 l.183-185, where snow-dominated catchments are defined as having a snow fraction larger than 0.3. On the other hand, the authors assessed groundwater by the BFI in a continuous way, so without a classification between groundwater-dominated and not groundwater-dominated catchments. I would suggest to do the same for both snow and groundwater, so either a continuous or a binary classification. Related to this, I think the authors can do a bit more to clarify the role of groundwater. On p.13 l.287-291, they state that soil properties and geology are important, but that these characteristics are commonly not available. However, the CAMELS-CL dataset does include information on these variables (and BFI) and I strongly encourage the authors to include this in their analysis. How does the BFI based on modelled data relate to the BFI and soil and geological variables of CAMELS-CL? The difference between snow and groundwater storage should also be discussed more clearly in the Discussion section. On p.17 l.368-370, the authors now mention both groundwater and snow, but it is unclear to which catchments they are referring and whether both types of storages might occur in the same catchment. This is important when they draw conclusions about the drivers for changes in RR relationships (for example on p.18 l.393-398), because the drivers are interrelated, like

they also mention in the Conclusion (l.426-429). So, in a revised manuscript I would like to see a clearer discussion of the role of these two drivers and their interrelations.

There is some unclarity about the period of the megadrought period investigated. Multiple years are mentioned, especially for the end year (2018/19/now). It is important to clarify this. It is currently for example, not clear which period your 8-yr average refers to. If you used data from 2010 to 2018, then this is a 9-year period. Different years / periods are also mentioned on, for example, l.11, 55, 128-129, 143, 150, 339, 417, 421. And in Fig.8 the MD seems to be 2007-2018. Please clarify the time period of the MD itself and which period of record you used for the analysis of the MD (I understand that these can be different).

The Discussion section needs to include a paragraph on the uncertainties in the HBV model and the classification decisions and how both of these could have influenced the results. As mentioned before, also a paragraph on anthropogenic influences should be added. I also suggest the authors to relate their work to that of Stoelzle et al. (2014).

I don't fully agree with the conceptualization of Carey et al. (2010) that catchments with lower storage are more resilient. You could argue that the opposite is true and those catchments are less resilient because they dry up immediately. Maybe the authors can add their view on this.

**Specific comments**
l.115-118: Please clarify how the distribution between semi-arid catchments was chosen. Was this based on literature, a random split, or an iterative analysis? Would your discussion of the results have been different if you would have chosen this split differently?

l.157: Please clarify how the two boxes were configured in HBV: in parallel or consecutive, and explain what the boxes represent and why this matches the situation in the catchments.

l.164-167: I'm not convinced that HBV can correctly distinguish between soil and groundwater storage if it does not represent groundwater correctly (as explained by the authors due to the lack of drainage when streamflow ceases and because there is no surface runoff in the model so all precipitation excess and snow melt move through the soil and groundwater boxes).

l.185: Why is 0.3 chosen to classify snow-dominated catchments? From literature? Have you done a sensitivity analysis on this number?

l.187: I'm assuming that you used modelled streamflow to calculate BFI?

l.279: High model efficiencies are not a surprise in snow-dominated catchments with a clear seasonal regime. Please discuss this.

Fig.7: It would be interesting to explore a bit more the catchments that deviate from the pattern, for example the ones north of 36 degrees that have lower correlations then their neighboring catchments. Would this be related to effects of groundwater or anthropogenic influences? Please explore.

l.349-351: Here you mention ET as an important factor. This is an important point that could be highlighted more, especially since ET during the recent MD is probably larger than during earlier 8-yr periods because of climate change. This should be discussed more in the Discussion section. I was also

wondering why you see the effect of ET only in the temperate catchments and not in the semi-arid ones? These would correspond more to the Australian catchments Saft used in her analysis.

l.358-362: Fig.9 needs more explanation in the text. The results presented in Fig.9 are now only mentioned in the Discussion section (l.399-404). This should be moved to the Results section. Also, how have the three cases been defined, per catchment or overall? I'm wondering whether the drought/wet years are actually the same in central and south Chile.

l.435-436: These catchments also have less ET.

**Technical corrections**
Would the title not be better as "during a multi-year drought" (singular) as the plural suggests that multiple drought periods like the MD were analysed.

I would suggest to check the English throughout the manuscript. There are quite a lot of issues with prepositions, for example on l.113-117, 174, 193, 234, 237, 325.

l.24: dependant > dependent

l.20-23: add latitudes to central and southern Chile so that the abstract is understandable on a standalone basis. Also, when you talk about snow-dominated or semi-arid regions, please add where they are located.

l.43: why "potential" drought propagation?

l.56: what do you mean with "global changes"?

l.108: "less than 100 mm **in** the north"

l. 129: "we compared them…" > what do you mean with "them"? Do you mean a value for each catchment?

l.143: 1018 > 2018

l.163: simulated > simulate

l.181: remove "then"

l.195 & 338-339: catchments memory > catchment memory

l.200: 1979-**2018**

l.201: **are** evident

Fig.2: the non-linear y-axis is confusing. You mention in the caption that each line is a catchment, but it would be helpful to indicate this on the figure axis as well.

l.223: what do you mean with "up to 90% of streamflow deficits"? Do the extremely dry years have streamflow deficits up to 90% or do 90% of streamflow deficits classify as extremely dry? Explain more clearly in the Methods section how you have calculated the deficit in % (% of what?).

Fig.3: So the boxplots exclude the MD, so only cover the period 1979-2009? Please add this in the caption.

l.233: "presenting **8-year** mean runoff"

l.246: "negative shift; that is, …"

l.278-279: "values of 0.72, …" > values of what? Nash-Sutcliffe model efficiency?

Fig.7: Are bar-plots the best way to visualize this? I'm assuming that the blue-grey bars are when the blue ones are below the grey, but what about the green bars then? Why not just use points?

l.319: should "from the previous year" be moved forwards to after "the precipitation"? Precipitation does not influence previous year streamflow, does it?

l.330: "the difference between runoff and precipitation deficits" on a yearly basis?

Fig.8: how have dry years been defined?

l.361: persistent > multiple (two and more)?

Fig.9: Panel a > a) and b), panel b > c) & d)

l.403: result > results

l.449: dependant > dependent

l.453: catchments response > catchment response

**References:**

Carey, S. K., Tetzlaff, D., Seibert, J., Soulsby, C., Buttle, J., Laudon, H., McDonnell, J., McGuire, K., Caissie, D., Shanley, J., Kennedy, M., Devito, K. and Pomeroy, J. W.: Inter-comparison of hydro-climatic regimes across northern catchments: Synchronicity, resistance and resilience, Hydrol. Process., 24(24), 3591–3602, doi:10.1002/hyp.7880, 2010.

Stoelzle, M., Stahl, K., Morhard, A., and Weiler, M. (2014), Streamflow sensitivity to drought scenarios in catchments with different geology, *Geophys. Res. Lett.*, 41, 6174– 6183, doi:10.1002/2014GL061344.

---

## Referee Comment (RC2) · Gemma Coxon (Referee) · 6 Aug 2020

**Review of 'Progressive water deficits using multi-year droughts in Central-Southern Chile' by Alvarez-Garreton et al.**

This paper tackles an interesting topic of multi-year droughts in Chile using a large-sample dataset CAMELS-CL. It first analyses observed streamflow and precipitation data to identify shifts in rainfall-runoff behaviour during the megadrought and then explores the role of hydrological memory in controlling drought propagation intensity using simulations from HBV.

Overall I enjoyed reading this paper – the figures are very well presented, it tackles an interesting topic and there are a range of analyses to support the conclusion. However, the paper needs to discuss in more detail the limitations of the study, greater clarity is needed in the methods section and the key messages of the paper could be better highlighted. Below are my suggestions and comments, I have a couple of general (more major) comments and then a series of minor comments.

**General Comments**

**Limitations.** I was missing a broader discussion of the limitations of the study in the discussion section. The authors mention the absence of physical factors in the analysis (L390-392) but the authors should also discuss

(1) *human impacts in these catchments* – I appreciate this study focuses on physical processes but humans can also play a large part in the intensification of drought propagation. Consequently, it is important to state how human-impacted these catchments are and whether human activities increased during the mega drought (particularly if groundwater abstractions increased because of deficits in streamflow).

(2) *evapotranspiration* – there is a brief analysis of ET in Section 4.4, but a limitation of this study is the absence of any detailed analyses on ET or temperature. Given their role for snow processes and intensification of drought, this is an important missing process from the analysis. The figure in the supplementary info showed high ET anomalies for the semi-arid catchments which had large RR shifts.

**HBV Modelling.** Given the richness of the CAMELS-CL dataset, I was a little surprised that the authors went down the route of extending the analysis with a model (as great as HBV is!) rather than further exploring the meteorological (e.g. ET), physical (e.g. soils, geology, land cover) and human impact characteristics of these catchments to explain the R-R shift in the megadrought. You could have used BFI from CAMELS-CL for Figure 6 and calculated baseflow contributions from the observed flow time series for the analysis in Figure 7. The authors need to:

(1) better clarify the contribution of the modelling to the paper

(2) better clarify where outputs from HBV are used in the analysis, particularly where *observed* streamflow or *modelled* streamflow is used throughout the text.

(3) comment more broadly on the uncertainties and limitations of HBV in the discussion

**Classifications.** There are lots of different classifications of catchments (snow-dominated/pluvial basins in Figure 5, semi-arid/temperate in Figure 8) which is very confusing for the reader. I would use one classification and keep it consistent throughout the paper.

**Minor Comments (in order of appearance in paper)**

**Abstract L16.** "mediated by groundwater flows" – not just groundwater flows but surely storage in snow packs too?

**Section 2 L89-90.** There are multiple rainfall and PET products available in CAMELS-CL, it would be helpful to specify exactly which hydrometeorological data products you used from the dataset.

**Section 2 L92-97.** The infilling of the flow time series is currently not clear in the paper and I have a number of questions related to this.

CAMELS-CL provides daily flow timeseries (as far as I am aware), which I then assume you aggregate up to monthly and then annual values for the rest of the analysis. So how do you define a month where there is no streamflow data – does this mean all days from that month are missing or a threshold of xx days? Do you calculate a sum of the daily flow in that month or take a mean (a mean would be less sensitive to missing data)?

Why did you select gauges with 15 years of data, is this really sufficient coverage of the period 1979-2018? What was the highest number of months that needed to be gap filled for a single station?

**Section 2 L108** change to **"**from less than 100 mm to the north **to more than 3000 mm** in the .."

**Figure 1.** Given the focus on groundwater dynamics you need a map of geology in Figure 1 and a description of the geology of Chile in the study description.

**Figure 1 Caption**. You need to add the source of the precipitation and temperature value either into the figure caption or the text.

**Section 3.1 L128.** Was there a reason for choosing 8 years to calculate the mean flux?

**Figure 3.** The red dots are quite hard to see – it may be worth increasing the size of the red dot or perhaps trying a red cross instead?

**Figure 6.** I was a little confused how Figure 6 was created – is this an average from all the snow-dominated and pluvial basins? Are Q, BF and snowmelt derived from HBV for this plot?

**Section 4.3.** You used HBV to calculate the baseflow index. Exactly how did you calculate the baseflow index and how does this value of baseflow index differ to the baseflow index calculated from observed streamflow and provided in CAMELS-CL.

**Figure 7**. I found the overlapping bars in Figure 7 quite confusing to interpret – have you thought about an alternative way to visualise these results as currently Figure 7a and 7b are quite difficult to interpret. Also is this seasonal runoff and baseflow analysed over the whole time period (i.e. from 1979-2010)?

**Section 4.4 L329.** "Figure 8 presents drought propagation for 25 years with negative anomalies over the last four decades". Do you mean negative anomalies in P and/or Q deficits? Or negative anomalies in the difference between runoff and precipitation deficits? If it is the latter then not all those years have negative anomalies (i.e. there are some years where the red dot is above 0%).

**Figure 9.** The text in L358-362 is essentially just a description of the figure and there should be some analysis of the results of Figure 9 here (currently, the analysis of the results is confusingly mixed into the discussion). The figure caption also needs to be more informative i.e. it refers to panel a and b

but there are subplots a-d in the plot, not really clear what Q-Pred R-R is or the number of cases or what '1', '2', '3' and '4' relate to on the x-axis of Figure 9b and 9d.

**Discussion L370.** Do you mean baseflow contribution to runoff – rather than low flow?

**Discussion L397.** What do you mean by "higher resistance"?

**Conclusions.** There is a lot of different analyses in this paper but the key message and results tend to get a little lost. I suggest to significantly shorten the conclusions – there is a lot of repetition in this section and removing it would better highlight the core results.

**Supplementary Material.** Figure S1 needs a figure caption.

---

## Author Comment (AC1) · 25 Sep 2020

The comment was uploaded in the form of a supplement:
https://hess.copernicus.org/preprints/hess-2020-249/hess-2020-249-AC1-
supplement.pdf

---

## Author Comment (AC2) · 25 Sep 2020

"Progressive water deficits during multi-year droughts in central-south Chile" by Alvarez-Garreton et al.

Response to Anne Van Loon and Gemma Coxon

We thank the two Referees, Anne Van Loon and Gemma Coxon, for their constructive comments on our paper.

The main changes proposed for the revised manuscript, which we think will improve the scientific approach and will help to clarify the main insights from the work, follow both referees' suggestions and include:

1) A new section with a classification scheme that groups the catchments based on their main hydrological regimes. With this procedure, we replace the formerly "semi-arid" and "temperate" basins by two, more objectively defined, new groups: snow-dominated and pluvial basins.

2) A deeper discussion regarding the potential influence of anthropic activities on the R-R shifts and overall interpretation of results. Major anthropic activities (e.g., agriculture, allocated water rights) will be accounted for in the revised Section 2: Study area and data. Based on this analysis, and in order to enable a more robust interpretation of results, we have filtered out 14 basins having reservoirs within.

3) A discussion on the HBV model limitations and uncertainties.

Additionally, we have updated the hydro-meteorological datasets until March 2020, which allows us to include the extreme dry conditions in central-south Chile in 2019 (one of driest year in a century).

Below we provide our replies to each Referee comment. For clarity, Referee comments are given in blue text, our responses are given in plain text and the proposed paper modifications in italics.

**Referee #1, Anne Van Loon:**

This paper uses a recently published dataset to look at changes in rainfall-runoff relationships during multi-year drought covering a range of different conditions in Chile. The authors conclude that both groundwater and snow play an important role and that some regions are more affected by prolonged drought and others by short extreme drought. The topic of the paper is important and the range of catchments included in the dataset provides interesting insights. The analysis is done well and the paper is generally well-written. The paper can be suitable for publication in HESS after a number of (relatively minor) revisions. Below I provide my suggestions for improvement of the paper.

We thank the reviewer for her positive comments.

**General comments**

1) The authors do not mention potential human influence in the catchments (except in the final sentence of the manuscript, l.461). It is unclear whether these catchments are completely without any human influence on the hydrology (which I doubt because they are quite large and cover a large part of Chile, so they are unlikely not to include reservoirs, forestry plantations, agriculture). There might for example be an increase of abstraction or change of land use during multi-year drought that might influence the rainfall-runoff relationship. CAMELS-CL includes information on land use, intervention degree, water rights and it would be greatly improve the paper if this was included in the analysis. If this cannot be included in the current paper, the authors must at least mention the degree of human influence in the catchments at the start of the paper and discuss potential effects of human influences on the results at the end (in the Discussion section).

We agree with the Referee at this point. In addition to catchment runoff mechanisms and the unprecedented dry conditions experienced within the megadrought (MD), another factor that may be influencing rainfall-runoff relationships are the anthropic activities within the catchments. Local activities may be particularly important during a multi-year drought, where the total available water within a catchment decreases, while the water consumption related to human activities (forestry, irrigated agriculture, etc) most probably not. In fact, the influence of reservoirs on catchment response may be higher during droughts than in normal years, given that irrigation must supply a larger portion of the water demanded by plants. This may contribute to a different rainfall-runoff response, when compared to the historical (pre-MD) catchment response.

Quantifying the influence of anthropic factors on water availability, particularly on the observed progressive water deficits during the MD, is part of the ongoing objectives from our research group. However, to address such analyses we require some additional datasets to characterise anthropic activities during and before the MD. These datasets are currently being implemented, but that will be available in ~1 year according to our planning. Such datasets include (i) land cover maps for different periods of time to enable the quantification of land use and land cover changes within the territory (note that the CAMELS-CL land cover data correspond to a single year, 2016); (ii) the quantification of water consumption from the different economic sectors associated to the generated land use maps, i.e., 'actual' water use time series; and (iii) the temporal time series of granted water use rights, i.e., 'potential' water use time series (water use rights in CAMELS-CL dataset correspond to the accumulated granted volumes in 2018).

Based on the above, we consider that a robust analysis of the anthropic influence on drought propagation during the MD, compared to pre-MD conditions, is beyond the scope of the current paper. Notwithstanding this, and by following the Referee suggestions, in the revised manuscript we will:

- Highlight this topic in the introduction
- Provide a description of the current state (i.e., during the MD) of human activities within the catchments based on land cover, water rights and reservoirs datasets from CAMELS-CL (revised Sect. 2: Study region and data).
- Provide a discussion (Sect. 5) about the potential effects of human influences on our findings, and the limitations of the available datasets to assess such influence. Despite these limitations, in order to enable a more robust interpretation of results, we will filter out 14 basins with reservoirs from the study domain.

2) The authors should be more specific about the role of snow vs. groundwater when they talk about memory (for example in the abstract l.16-18). This starts with the classification of catchments in semi-arid vs. temperate catchments, which seems to be partly overlapping with snow-dominated vs. rainfall- dominated catchments but this is not clearly identified. For example, on p.5 l.111-113 the authors state that in central Chile there is an Mediterranean regime, whereas in snow-dominated basins "streamflow peaks in spring / early summer", which makes me think that the snow-dominated basins are located in southern Chile. But then the classification (l.113-118) seems to be only based on precipitation and not on snow and Fig.1 shows that there is more snow in the north. Also, an additional classification of catchments is introduced on p.7 l.183-185, where snow-dominated catchments are defined as having a snow fraction larger than 0.3. On the other hand, the authors assessed groundwater by the BFI in a continuous way, so without a classification between groundwater-dominated and not groundwater- dominated catchments. I would suggest to do the same for both snow and groundwater, so either a continuous or a binary classification.

Both Referees have commented on the way of characterising the study catchments, pointing out the confusion of using different classification schemes, such as 'semi-arid and Mediterranean basins' (based on total precipitation), and 'snow-dominated and pluvial basins' (based on snow fraction). Further, in the submitted manuscript we also assessed the role of groundwater (GW) based on a BFI computed from HBV, which can be seen as another classification scheme.

Addressing these comments, in the revised manuscript we will provide a unique classification scheme to group the basins by their main hydrological regimes, based on k-means clustering algorithm (Kanungo et al., 2002). To implement this classification, we applied k-means clustering to three key hydro-meteorological basin features (normalised variables) provided in Section 2 and summarised in the following table:

| Variable name | Description (all variables are computed at the catchment scale) |
|---|---|
| Pw_Pa_clim | Ratio of mean fall-winter precipitation (april to september) to mean annual precipitation (period 1979-2018) from CAMELS-CL dataset |
| Qw_Qa_clim | Ratio of mean fall-winter streamflow (april to september) to mean annual streamflow (period 1979-2018) from CAMELS-CL dataset |
| Snow fraction | Long-term snow fraction, computed as the average of ERA5-L solid to total precipitation ratio for the period 1981-2018 |

Summary of classification results:

[Figure]

Cluster 1 (40 basins):
Proposed name: Snow-dominated
Main features: Higher snow fraction, higher streamflow summer seasonality (Qw_Qa_clim very low), higher precipitation winter seasonality (Pw_Pa_clim).

Cluster 1 (70 basins):
Proposed name: Pluvial
Main features: Lower snow fraction, higher winter streamflow seasonality (large Qw_Qa_clim values), lower winter precipitation seasonality (Pw_Pa_clim).

This revised nomenclature will be used across the complete manuscript. The classification procedure and the cluster results will be added as part of the study area description, in Sect. 2. The rest of the Figures from the manuscript will be reordered in order to keep consistency with this classification.

Below we provide the locations of the basins from the two clusters:

[Figure]

Reference: T. Kanungo, D. M. Mount, N. S. Netanyahu, C. D. Piatko, R. Silverman and A. Y. Wu, "An efficient k-means clustering algorithm: analysis and implementation," in IEEE Transactions on Pattern Analysis and Machine Intelligence, vol. 24, no. 7, pp. 881-892, July 2002, doi: 10.1109/TPAMI.2002.1017616.

3) Related to this, I think the authors can do a bit more to clarify the role of groundwater. On p.13 l.287-291, they state that soil properties and geology are important, but that these characteristics are commonly not available. However, the CAMELS-CL dataset does include information on these variables (and BFI) and I strongly encourage

the authors to include this in their analysis. How does the BFI based on modelled data relate to the BFI and soil and geological variables of CAMELS-CL?

Before addressing this comment, and by considering the comments related to BFI from Referee 2 as well, we want to emphasise that the modelled 'BFI' we computed from HBV is different from the BFI from CAMELS-CL:

1) The BFI from CAMELS-CL dataset was computed as the ratio of mean daily baseflow to mean daily discharge. Hydrograph separation to compute daily baseflow was done by using Ladson et al. (2013) digital filter with α set to 0.925 (see Table 3, Alvarez-Garreton et al., 2018). The baseflow conceptualization refers to the dynamic components of the total runoff, where baseflow is the slow response to precipitation, while the quick flow (total flow minus baseflow) is the fast response to precipitation. This separation approach focuses on the response timing of an event; however, it does not explicitly indicate the source of the water (e.g., surface, shallow soil or deep soil).

2) The BFI from HBV presented in the manuscript was computed in this paper as the mean annual runoff from the lower GW reservoir, normalised by the mean annual runoff. By contrast to the BFI from hydrograph separation, the index computed from HBV represents the slow groundwater contribution to total streamflow. For this work, HBV was implemented with two soil boxes, the upper soil box represents faster groundwater release to total streamflow, and the lower soil box represents a slower groundwater release to total streamflow.

We realise that this different conceptualisations and computation of both BFI indices (which are not directly comparable), lead to confusion. To address this, and to emphasise the information provided by HBV groundwater boxes, in the revised manuscript we will name the HBV groundwater index (former BFI) as "GWI", which corresponds to the mean annual water coming groundwater at slower rates normalised by the mean annual streamflow.

Also, by following the Referee suggestion, in the revised manuscript we will include the BFI derived from CAMELS-CL in Section 2 (Study area and data). The text will be corrected throughout the manuscript. In addition, we will add a new section presenting the indices from HBV and the model performance, and how these relate with the two main catchment groups (snow-dominated and pluvial). The new Results Section will be named "4.3 HBV simulations and catchment runoff mechanisms".

The address the specific question of how BFI from CAMELS-CL and GWI from HBV relate, in the following Figure we plot the correlation between these two variables (colours correspond to snow fraction from ERA5-L data, and marker size corresponds to the basin area):

[Figure]

The coefficient of determination between these two variables is $r^2 = 0.34$. Although the difference of BFI and GWI will be discussed in the revised manuscript, this Figure will not be incorporated, since we do not think it helps to address the research objectives of the paper.

4) The difference between snow and groundwater storage should also be discussed more clearly in the Discussion section. On p.17 l.368-370, the authors now mention both groundwater and snow, but it is unclear to which catchments they are referring and whether both types of storages might occur in the same catchment. This is

important when they draw conclusions about the drivers for changes in RR relationships (for example on p.18 l.393-398), because the drivers are interrelated, like they also mention in the Conclusion (l.426-429). So, in a revised manuscript I would like to see a clearer discussion of the role of these two drivers and their interrelations.

We think the new classification scheme will help to clarify this point since both storages are present in snow-dominated, while only GW storages are present in pluvial basins.

Catchments in the snow-dominated group feature stronger hydrological memory beyond one year, and larger R-R shifts, which translate to progressive water deficits during multi-year droughts. On the other hand, pluvial catchments feature weaker hydrological memory and lower R-R shifts.

We agree with the Referee in that the GW and snow storages are interrelated, and we illustrate this in Fig. 6a. This plot indicates that for larger snow fraction (i.e., where snow storage is more important and more markedly dominates the hydrologic regime of a catchment), the larger the GWI is (i.e., larger portion of the total runoff comes from the deep GW storage). Please note that in this revision process, we have renamed the former BFI as GWI, to differentiate the BFI coming from CAMELS-CL and based on daily streamflow observations and the GWI computed from the HBV model (See our reply to general comment 3). In the revised manuscript, we will add these comments and clarification to the Discussion and Conclusions sections.

5) There is some unclarity about the period of the megadrought period investigated. Multiple years are mentioned, especially for the end year (2018/19/now). It is important to clarify this. It is currently for example, not clear which period your 8-yr average refers to. If you used data from 2010 to 2018, then this is a 9-year period. Different years / periods are also mentioned on, for example, l.11, 55, 128-129, 143, 150, 339, 417, 421. And in Fig.8 the MD seems to be 2007-2018. Please clarify the time period of the MD itself and which period of record you used for the analysis of the MD (I understand that these can be different).

Previous studies have identified the start of the MD in 2010. This MD has been characterised by consecutive years with precipitation deficits over a wide extension on the national territory (central and south Chile). We adopted the same initial start year (albeit the MD starts earlier in some regions), and we analysed the data up to the available date, since the MD is still present in most of the territory.

In the revised manuscript, we will add the recent hydrological year spanning from April 2019 to March 2020. Hence, the analyses now will include the extremely dry 2019, which featured precipitation deficits up to 90% in central Chile. Therefore, the MD period in the revised manuscript will be 2010-2019. We will clarify this in the revised manuscript.

6) The Discussion section needs to include a paragraph on the uncertainties in the HBV model and the classification decisions and how both of these could have influenced the results. As mentioned before, also a paragraph on anthropogenic influences should be added. I also suggest the authors to relate their work to that of Stoelzle et al. (2014).

Discussion about the classification scheme and the anthropic influence based on our previous replies will be added to the revised manuscript.

Regarding the uncertainties in HBV, we will add the following text into the discussion section:

*As any hydrological modelling, our simulations using the HBV model are affected by different types of uncertainties including parameter uncertainty and model structure uncertainty (e.g. Uhlenbrook et al., 1999) and observation uncertainties (MicMillan et al., 2018). The HBV model with its simple structure does only allow for limited long-term memory effects. Actually, this is an interesting point of this study as simulation residuals can be interpreted as a consequence of this lack of more complex long-term memory effects.*

*McMillan, HK, Westerberg, IK, Krueger, T. Hydrological data uncertainty and its implications. WIREs Water. 2018; 5:e1319. https://doi.org/10.1002/wat2.1319*

*Uhlenbrook, S., Seibert, J., Leibundgut, Ch. and Rodhe, A., 1999, Prediction uncertainty of conceptual rainfall-runoff models caused by problems to identify model parameters and structure, Hydrological Sciences - Journal des Sciences Hydrologiques 44(5): 779-798.*

7) I don't fully agree with the conceptualization of Carey et al. (2010) that catchments with lower storage are more resilient. You could argue that the opposite is true and those catchments are less resilient because they dry up immediately. Maybe the authors can add their view on this.

The concepts of 'resistance' and 'resilience' in Carey et al. (2010) are adopted from ecology, and may indeed be different to what we would interpret in hydrology. Here we provide some context on the conceptualisation of Carey et al. (2010), followed by our view on this.

From Carey et al. (2010):

"we define two functional traits of catchments taken from ecology: resistance and resilience (Folke et al., 2004; Potts et al., 2006). From a catchment perspective, resistance measures the degree to which runoff is coupled/synchronized with precipitation. Catchments that can store water over long time periods (months or years) and release water gradually to the stream have a high resistance, whereas catchments that systematically transfer precipitation into discharge have low resistance"

"Resilience measures the degree to which a catchment can adjust to normal functioning following perturbations from events such as drought or extreme precipitation. It is hypothesized that catchments with high resilience are able to sustain their expected precipitation–discharge relations in the light of changing inputs, whereas catchments with low resilience are sensitive to changes in inputs and exhibit enhanced threshold response behaviour."

"Catchments such as Strontian and HJ Andrews, where P >> S, have a higher resilience (P = precipitation, S = basin storage). This is particularly the case for Strontian where very low S (steep topography and thin soils) indicates that its functional relation between P and Q is insensitive to change. Conversely, catchments with low corr(Qmo,Pmo), higher S and lower P may exhibit decreased resilience, as changes in snow storage and soil storage can strongly impact the ability of the catchment to generate runoff".

Our view: The concept of 'resistance' defined by Carey et al. (2010) may be related to catchment hydrological memory, however, their concept of resilience may not be directly related to our approach. Relating the definition from Carey et al., (2010) with our analyses, a resilient catchment would be less prone to feature a shift in R-R relationship during the MD, and this would be the case for those basins where P>>S (i.e., P/S >>1), not necessarily where S is low. In fact, a catchment with low S can dry up quicker than others, as pointed out by the Referee, and this in turn may generate higher deficits during prolonged precipitation deficits. In order to highlight this, in the revised manuscript we will specify that the resilience conceptualisation from Carey et al. (2010) is based on P and S, and not only on the values of S. In this sense, our catchments featuring larger R-R shifts during the MD are related to a lower resilience, probably since P is very low, which does not mean that lower S is associated with higher resilience.

**Specific comments**

l.115-118: Please clarify how the distribution between semi-arid catchments was chosen. Was this based on literature, a random split, or an iterative analysis? Would your discussion of the results have been different if you would have chosen this split differently?

The initial catchment classification was based on typical regime differentiation in Chile, due the large rainfall gradient from north to south, but indeed this simple split (semi-arid & rainy) is not really the best choice regarding the purposes of this study. That's why now we use a classification scheme for snow vs. pluvial regime identification. Please see our reply to the general comment 2.

l.157: Please clarify how the two boxes were configured in HBV: in parallel or consecutive, and explain what the boxes represent and why this matches the situation in the catchments.

The HBV model was configured by two consecutive boxes, representing the soil upper zone and the soil lower zone. Each timestep a certain amount of water is percolating from the soil upper zone to the soil lower zone. The soil lower zone has one linear outflow, representing a slow groundwater flow (could be associated to a baseflow component from a hydrograph separation approach), whereas the soil upper zone has two linear outflows,

representing intermediate and fast groundwater flow (could be associated to an intermediate and peak flows), of which the latter is only activated in case a certain water content threshold is exceeded.

l.164-167: I'm not convinced that HBV can correctly distinguish between soil and groundwater storage if it does not represent groundwater correctly (as explained by the authors due to the lack of drainage when streamflow ceases and because there is no surface runoff in the model so all precipitation excess and snow melt move through the soil and groundwater boxes).

Despite the limitations regarding HBV (or any other model), we do not think we can assess that HBV is not representing GW correctly. The model has fairly good performance across the different types of basins, the conceptualization of the model is representing the runoff mechanisms that are consistent with the hydrologic regimes we assessed from observations (i.e., from the classification scheme).

Despite the above limitations, the simulation of the runoff mechanisms from HBV allows us to further understand how water travels through the basins, how long does it take, and how these characteristics may relate to the intensification of drought propagation during the MD. We will add this discussion in the revised manuscript.

l.185: Why is 0.3 chosen to classify snow-dominated catchments? From literature? Have you done a sensitivity analysis on this number?

This is now part of the classification scheme. Snow-dominated basins are now characterised by high values of snow fraction and other regime features (precipitation and streamflow seasonality). Please see our reply to general comment 2).

l.187: I'm assuming that you used modelled streamflow to calculate BFI?

Yes. But we have changed the name of the former BFI computation, to avoid confusion with the BFI from CAMELS-CL. This new nomenclature will be presented in the new section 4.3 ("HBV simulations and catchment runoff mechanisms"). Please refer to our reply to general comment 3.

l.279: High model efficiencies are not a surprise in snow-dominated catchments with a clear seasonal regime. Please discuss this.

The model efficiencies are computed for daily time series. We will add further discussion to model performance metrics in the new section 4.3. ("HBV simulations and catchment runoff mechanisms"). Please refer to our reply to general comment 3. Below we provide a plot showing the NPE and KGE for the two types of basins: snow-dominated and pluvial. This Figure shows that there are no significant differences in model performance among the pluvial and snow-dominated basins.

[Figure]

Fig.7: It would be interesting to explore a bit more the catchments that deviate from the pattern, for example the ones north of 36 degrees that have lower correlations then their neighboring catchments. Would this be related to effects of groundwater or anthropogenic influences? Please explore.

The basins of this figure will be re-ordered based on the group they belong to (snow-dominated, mixed or pluvial), thus the pattern will be not the same as the submitted manuscript. Nevertheless, we will explore a couple of basins deviating from the pattern.

l.349-351: Here you mention ET as an important factor. This is an important point that could be highlighted more, especially since ET during the recent MD is probably larger than during earlier 8-yr periods because of climate change. This should be discussed more in the Discussion section. I was also wondering why you see the effect of ET only in the temperate catchments and not in the semi-arid ones? These would correspond more to the Australian catchments Saft used in her analysis.

We agree with the Referee in that ET may contribute to the observed changes in catchment response during the MD. For similar precipitation deficits than the historic period (1979-2009), larger ET anomalies during the MD modulated by the higher temperatures for some years during the MD may lead to lower runoff generation, when compared to the historical period.

To include a deeper discussion on ET, we will add a third panel in Fig. 3 with the 8-year averages for ET within the study catchments. We will further discuss these results in the discussion as well.

l.358-362: Fig.9 needs more explanation in the text. The results presented in Fig.9 are now only mentioned in the Discussion section (l.399-404). This should be moved to the Results section. Also, how have the three cases been defined, per catchment or overall? I'm wondering whether the drought/wet years are actually the same in central and south Chile.

We will add further discussion on the results in Figure 9. In the submitted manuscript, the years of extreme droughts were selected with an overall approach, i.e., when conditions were met for more than half of the basins: mean precipitation anomalies below -50% in semi-arid basins and mean precipitation anomalies below -25% in temperate basins, preceded by a wet year (mean precipitation above average) or preceded by a dry year (mean precipitation below average).

As the Referee points out, there have been years where meteorological droughts are not experienced within the complete domain, as can be seen from Figure 2a and 2c (e.g., for the years 1979, 1981, 1983, 1994, 1995). After 1996 however, most meteorological deficits have been observed throughout the complete study domain.

The Figure will be modified based on the new classification scheme.

l.435-436: These catchments also have less ET.

Pluvial basins in southern Chile are generally wetter and then have more ET, but a lower evaporative fraction (ET/P). This will be added in the indicated text.

**Technical corrections**

Would the title not be better as "during a multi-year drought" (singular) as the plural suggests that multiple drought periods like the MD were analysed.

While the R-R shifts were analysed during the MD, the extremes vs persistent droughts (section 4.5) analysed other years and periods with consecutive precipitation deficits, not only the MD. In this way, our conclusions relating hydrological memory with progressive water deficits or intensification of drought propagation, are drawn for the general case of consecutive precipitation deficits, despite the specific insights we gained from the MD period.

I would suggest to check the English throughout the manuscript. There are quite a lot of issues with prepositions, for example on l.113-117, 174, 193, 234, 237, 325.

The complete manuscript will be proof read to correct typos and grammar issues.

l.24: dependant > dependent

Corrected.

l.20-23: add latitudes to central and southern Chile so that the abstract is understandable on a standalone basis. Also, when you talk about snow-dominated or semi-arid regions, please add where they are located.

Latitudes will be added to the abstract. The grouped basins will be defined by classification (see our reply to general comment 2), and their locations will be provided in the abstract.

l.43: why "potential" drought propagation?

Deleted from text.

l.56: what do you mean with "global changes"?

We used the term global changes to refer to changes not only related to global warming, but also to other anthropic-related changes, such as land cover land use changes, intensification of irrigated agriculture, etc.

l.108: "less than 100 mm in the north"

Corrected.

l. 129: "we compared them..." > what do you mean with "them"? Do you mean a value for each catchment?

Corrected. "We compared the 8-year mean values during the MD with historical eight-year average values"

l.143: 1018 > 2018

Corrected.

l.163: simulated > simulate

Corrected.

l.181: remove "then"

Corrected.

l.195 & 338-339: catchments memory > catchment memory

Corrected.

l.200: 1979-2018

Corrected.

l.201: are evident

Corrected.

Fig.2: the non-linear y-axis is confusing. You mention in the caption that each line is a catchment, but it would be helpful to indicate this on the figure axis as well.

We will clarify this in the Figure axis. Additionally, the catchments will be ordered now by their classification groups, and not by latitude.

l.223: what do you mean with "up to 90% of streamflow deficits"? Do the extremely dry years have streamflow deficits up to 90% or do 90% of streamflow deficits classify as extremely dry? Explain more clearly in the Methods section how you have calculated the deficit in % (% of what?).

We will rewrite the sentence as *"(up to 90% of streamflow deficits with a median of 57%)"*.

In Section 3.1 we explained that relative anomalies were computed as: "We computed relative anomalies of hydrological years (April-March) as deviations from climatological means (period 1979-2009, i.e., excluding the MD), normalised by the climatological mean of each time series." We will add the following equation to avoid confusion:

Qa_anom = 100* (Qa - Qclim) / Qclim, where Qa is the annual streamflow and Qclim is the mean annual streamflow for the period 1979-2009.

Fig.3: So the boxplots exclude the MD, so only cover the period 1979-2009? Please add this in the caption.

We will add this to the Figure caption.

l.233: "presenting 8-year mean runoff"

Corrected.

l.246: "negative shift; that is, ..."

Corrected.

l.278-279: "values of 0.72, ..." > values of what? Nash-Sutcliffe model efficiency?

The performance statistic we used for calibration was the non-parametric variation of Kling-Gupta efficiency, NPE. This will be added to the text.

Fig.7: Are bar-plots the best way to visualize this? I'm assuming that the blue-grey bars are when the blue ones are below the grey, but what about the green bars then? Why not just use points?

We will try different points/bar combinations in order to help visualisation in this figure.

l.319: should "from the previous year" be moved forwards to after "the precipitation"? Precipitation does not influence previous year streamflow, does it?

Yes. This was a mistake. We will correct the text accordingly.

l.330: "the difference between runoff and precipitation deficits" on a yearly basis? Fig.8: how have dry years been defined?

Yes, yearly basis. We will add this to the text. In the submitted manuscript, years were selected when mean precipitation deficits were larger than 25% for any of the two basins groups over the last four decades

In order to better visualise the drought propagation and recovery, we will replace the discretisation in Fig. 8 by a plot with continuous years, as the Figure below, where average annual anomalies of precipitation and streamflow for the snow-dominated basins and pluvial basins are presented. The area between the two curves is coloured in blue when the streamflow anomaly is larger than the precipitation anomaly, and in red if not. This Figure, along with the corresponding Caption and explanation, will replace Figure 8.

[Figure]

a) Snow–dominated catchments

b) Pluvial catchments

l.361: persistent > multiple (two and more)?

We will specify the range considered in this case: *"iii) persistent years (2 up to 4 consecutive years) with precipitation anomalies below -5%."*

Fig.9: Panel a > a) and b), panel b > c) & d)

We will reorder the panels as suggested, and will also include the two types of catchments.

l.403: result > results

Corrected.

l.449: dependant > dependent

Corrected.

l.453: catchments response > catchment response

Corrected.

**References**

Carey, S. K., Tetzlaff, D., Seibert, J., Soulsby, C., Buttle, J., Laudon, H., McDonnell, J., McGuire, K., Caissie, D., Shanley, J., Kennedy, M., Devito, K. and Pomeroy, J. W.: Inter-comparison of hydro-climatic regimes across northern catchments: Synchronicity, resistance and resilience, Hydrol. Process., 24(24), 3591– 3602, doi:10.1002/hyp.7880, 2010.

Stoelzle, M., Stahl, K., Morhard, A., and Weiler, M. (2014), Streamflow sensitivity to drought scenarios in catchments with different geology, Geophys. Res. Lett., 41, 6174– 6183, doi:10.1002/2014GL061344.

**Referee #2, Gemma Coxon:**

This paper tackles an interesting topic of multi-year droughts in Chile using a large-sample dataset CAMELS-CL. It first analyses observed streamflow and precipitation data to identify shifts in rainfall-runoff behaviour during the megadrought and then explores the role of hydrological memory in controlling drought propagation intensity using simulations from HBV.

Overall I enjoyed reading this paper – the figures are very well presented, it tackles an interesting topic and there are a range of analyses to support the conclusion. However, the paper needs to discuss in more detail the limitations of the study, greater clarity is needed in the methods section and the key messages of the paper could be better highlighted. Below are my suggestions and comments, I have a couple of general (more major) comments and then a series of minor comments.

We thank the reviewer for her positive and constructive comments.

**General Comments**

1) Limitations. I was missing a broader discussion of the limitations of the study in the discussion section. The authors mention the absence of physical factors in the analysis (L390-392) but the authors should also discuss

(i) human impacts in these catchments – I appreciate this study focuses on physical processes but humans can also play a large part in the intensification of drought propagation. Consequently, it is important to state how human-impacted these catchments are and whether human activities increased during the mega drought (particularly if groundwater abstractions increased because of deficits in streamflow).

We agree with the Referee in that the local anthropic activities within the catchments may play an important role in drought propagation, and that the current manuscript does not deepen on this point. We have addressed this limitation in the revised manuscript, describing the current state of human activities within the basins, and highlighting the data limitations that define the scope of a robust anthropic analysis within the paper. Please refer to our reply to the general comment 1 from Referee 1.

(ii) evapotranspiration – there is a brief analysis of ET in Section 4.4, but a limitation of this study is the absence of any detailed analyses on ET or temperature. Given their role for snow processes and intensification of drought, this is an important missing process from the analysis. The figure in the supplementary info showed high ET anomalies for the semi-arid catchments which had large RR shifts.

We agree with the Referee in that ET may contribute to the observed changes in R-R response during the MD. For similar precipitation deficits than the historic period (1979-2009), larger ET anomalies during the MD modulated by the higher temperatures for some years during the MD may lead to lower runoff generation, when compared to the historical period.

Please see our reply to the Specific comments from Referee 1 where we discuss how we will address ET analysis in the revised manuscript.

2) HBV Modelling. Given the richness of the CAMELS-CL dataset, I was a little surprised that the authors went down the route of extending the analysis with a model (as great as HBV is!) rather than further exploring the meteorological (e.g. ET), physical (e.g. soils, geology, land cover) and human impact characteristics of these catchments to explain the R-R shift in the megadrought.

In addition to exploring drought propagation only by using the CAMELS-CL dataset, we decided to incorporate a hydrological model in order to address two main challenges of the analysis. Firstly, we aimed at improving our current understanding on runoff mechanisms from a process-based perspective. In particular, we aimed at assessing the role of snow and GW in runoff generation. Secondly, a process-based model overcomes some of the limitations of diagnosing the progressive water deficits only from the shift in annual R-R relationships. Recall that our hypothesis is that catchments with stronger hydrological memory (i.e., longer-than-a-year memory) feature progressive water deficits during multi-year droughts. In testing this hypothesis, a process-based model that is able to capture this memory should better simulate the observed progressive deficits, compared to the simulations from R-R regressions that only consider the precipitation of the same year.

Regarding the factors that the Referee suggest to further explore, we agree with all of them, and below we explain how each will be included in the revised manuscript:

(i) Meteorological data: the ET information will be further analysed in the revised manuscript (please see our previous reply).

(ii) Physical data (geology, soils and land cover): in the revised manuscript, we will provide a description of the catchment geological properties based on CAMELS-CL dataset. It should be noted that CAMELS-CL does not include soil properties. Land cover dataset are discussed in the following point (as part of human-related datasets)

(iii) Human-related datasets: As we explained in our reply to the general comment 1 from Referee 1, there are some limitations in the CAMELS-CL human-related datasets, related to the lack of land cover characteristics prior 2016, and the lack of human intervention characteristics prior the MD. In our opinion, these limitations should be overcome before we can attribute the R-R shifts to anthropic activities. Based on these limitations and on the scope of the paper, in the submitted manuscript we focused on the runoff mechanisms to understand the R-R shifts, which we hypothesise are among the key factors explaining the progressive water deficits during the MD. Notwithstanding this, in the revised manuscript we will incorporate a description of the current state of land cover and human intervention within the basins, as well as a discussion of their potential impacts on drought propagation.

You could have used BFI from CAMELS-CL for Figure 6 and calculated baseflow contributions from the observed flow time series for the analysis in Figure 7.

The Referee coincides with the first Referee regarding the use of BFI from CAMELS-CL in the analysis. In the revised manuscript, we will incorporate these data to the analysis (please see our reply to the general comment 3 from Referee 1).

The Referee is right, both figures relating baseflow to the hydrological memory (Fig. 6) and precipitation (Fig. 7b) could be generated based on the CAMELS-CL BFI and on the baseflow contributions computed from the digital filter applied to the observed streamflow in CAMELS-CL, respectively. However, we think that a process-based estimation of seasonal groundwater contribution to total runoff (i.e., GWI from HBV, which is the new nomenclature adopted to avoid confusion with BFI indices, as explained in our reply to the general comment 3 from Referee 1) is a better representation of the slow contribution from groundwater than the digital filter with a fixed α parameter used in CAMELS-CL. This in turn should be better related to the hydrological memory of a catchment.

The authors need to:

(i) better clarify the contribution of the modelling to the paper

In the revised manuscript, we will add the reasons for incorporating a process-based model to the analysis, as explained above.

(ii) better clarify where outputs from HBV are used in the analysis, particularly where observed streamflow or modelled streamflow is used throughout the text.

We will clarify where the outputs from HBV are used.

(iii) comment more broadly on the uncertainties and limitations of HBV in the discussion

We will add the suggested discussion. Please see our reply to general comment 6 from Referee 1.

3) Classifications. There are lots of different classifications of catchments (snow-dominated/pluvial basins in Figure 5, semi-arid/temperate in Figure 8) which is very confusing for the reader. I would use one classification and keep it consistent throughout the paper.

We agree with both Referees regarding the way the study catchments are classified, leading to confusion when different classifications are used. The revised manuscript will include a single snow/pluvial regime characterization based in an objective classification scheme. Please refer to our reply to the general comment 2 from Referee 1.

**Minor Comments (in order of appearance in paper)**

Abstract L16. "mediated by groundwater flows" – not just groundwater flows but surely storage in snow packs too?

Yes. This will be corrected in the revised manuscript.

Section 2 L89-90. There are multiple rainfall and PET products available in CAMELS-CL, it would be helpful to specify exactly which hydrometeorological data products you used from the dataset.

The Referee is right, this info was missing in the text. We have used the CR2MET product, which was updated until 2019 for this work. We will specify which dataset was used.

Section 2 L92-97. The infilling of the flow time series is currently not clear in the paper and I have a number of questions related to this. CAMELS-CL provides daily flow timeseries (as far as I am aware), which I then assume you aggregate up to monthly and then annual values for the rest of the analysis.

Yes, correct.

So how do you define a month where there is no streamflow data – does this mean all days from that month are missing or a threshold of xx days? Do you calculate a sum of the daily flow in that month or take a mean (a mean would be less sensitive to missing data)?

In CAMELS-CL, monthly streamflow values are calculated when 15 or more days per month have valid data. If >15 days are available, a mean monthly value is obtained as the average of the available daily streamflow. This mean monthly streamflow is then aggregated into the total number of days within the month to get total monthly runoff.

Why did you select gauges with 15 years of data, is this really sufficient coverage of the period 1979- 2018?

The 15 years threshold was chosen in order to have a minimum of points during the MD for the R-R shifts analysis. We filtered out those basins with: 1) less than 5 years of data within the MD and ii) less than 10 years of data before the MD. Although this criterion implies a minimum threshold of 15 years, the basins that fulfilled this criterion have more than 15 years of data. Below we provide the histograms of number of years with data for each basin, for the period before the MD (1979-2009, left panel) and during the MD (2010-2018, right panel):

[Figure]

[Figure]

In summary, 70% of the study catchments (87 out of 124) have 30 years of data in the historical pre-MD period. During the MD period, 77% of the catchments (87 out of 124) have 9 years of data. Thanks for pointing this out, we will clarify these results in the revised manuscript.

What was the highest number of months that needed to be gap filled for a single station?

The gap filling scheme includes several parameters, including a minimum number of valid observations, in this case set to 75% of the target period (1979-2019). That is, a station can be potentially filled up if all gaps do not exceed 25% of the period. Each missing month is then independently assessed based on a number (ensemble) of multivariate models that use covariant records from other stations as predictors. A given linear model is used if it shows a predictive power ($R^2$) larger than 0.75.

Section 2 L108 change to "from less than 100 mm to the north to more than 3000 mm in the .."

Corrected.

Figure 1. Given the focus on groundwater dynamics you need a map of geology in Figure 1 and a description of the geology of Chile in the study description.

Geology map will be added to Figure 1. And the geology of the study catchments will be discussed based on the clusters they belong to.

Figure 1 Caption. You need to add the source of the precipitation and temperature value either into the figure caption or the text.

Added.

Section 3.1 L128. Was there a reason for choosing 8 years to calculate the mean flux?

We chose this number to be conservative with the missing streamflow data during the MD. However, by arguing that 77% of the basins have 9 years of data during the MD (see our histograms in the previous reply), we will change this into 9-year mean values (i.e., periods equally long as the MD).

Figure 3. The red dots are quite hard to see – it may be worth increasing the size of the red dot or perhaps trying a red cross instead?

We will try a different configuration in the revised manuscript to improve visualisation.

Figure 6. I was a little confused how Figure 6 was created – is this an average from all the snow- dominated and pluvial basins? Are Q, BF and snowmelt derived from HBV for this plot?

Fig. 6 shows all the catchments together. We will clarify which variables come from the HBV model and which ones from observations.

Section 4.3. You used HBV to calculate the baseflow index. Exactly how did you calculate the baseflow index and how does this value of baseflow index differ to the baseflow index calculated from observed streamflow and provided in CAMELS-CL.

Please see our reply to general comment 3 from Referee 1 where these questions are addressed.

Figure 7. I found the overlapping bars in Figure 7 quite confusing to interpret – have you thought about an alternative way to visualise these results as currently Figure 7a and 7b are quite difficult to interpret. Also is this seasonal runoff and baseflow analysed over the whole time period (i.e. from 1979-2010)?

We will try different plot combinations in order to help visualise this figure.

Seasonal correlations were computed for the complete period of record (1979-2019 in the revised manuscript), i.e., including the MD. This will be clarified in the revised manuscript.

Section 4.4 L329. "Figure 8 presents drought propagation for 25 years with negative anomalies over the last four decades". Do you mean negative anomalies in P and/or Q deficits? Or negative anomalies in the difference between runoff and precipitation deficits? If it is the latter then not all those years have negative anomalies (i.e. there are some years where the red dot is above 0%).

We meant negative anomalies in precipitation, although this was not very clear in the text. We decided to replace Fig. 8 by time series plots showing precipitation and streamflow anomalies for the snow-dominated and pluvial basins. Please see our reply to technical correction from Referee 1.

[Figure]

Figure 9. The text in L358-362 is essentially just a description of the figure and there should be some analysis of the results of Figure 9 here (currently, the analysis of the results is confusingly mixed into the discussion). The figure caption also needs to be more informative i.e. it refers to panel a and b but there are subplots a-d in the plot, not really clear what Q-Pred R-R is or the number of cases or what '1', '2', '3' and '4' relate to on the x-axis of Figure 9b and 9d.

This comment is in line with Referee 1 regarding the analysis of this Figure. We will move the description of results from the Discussion into the Results Section. Captions will be corrected.

Discussion L370. Do you mean baseflow contribution to runoff – rather than low flow? Discussion L397. What do you mean by "higher resistance"?

The concept of resistance was included in the discussion of Carey et al., paper. We have clarified this point in our reply to general comment 7 from Referee 1.

Conclusions. There is a lot of different analyses in this paper but the key message and results tend to get a little lost. I suggest to significantly shorten the conclusions – there is a lot of repetition in this section and removing it would better highlight the core results.

The conclusion section will be condensed, deleting repetitions.

Supplementary Material. Figure S1 needs a figure caption.

Caption will be added.

---

## Author Response (AR2)

"Progressive water deficits during multiyear droughts in basins with long hydrological memory in Chile" by Alvarez-Garreton et al.

Response to Gemma Coxon

We thank the referee Gemma Coxon for her constructive comments on our paper.

Below we provide our replies to each Referee comment. For clarity, Referee comments are given in blue text, our responses are given in plain text and the proposed paper modifications in italics.

The authors have carefully considered and responded to all my comments. They have made substantial changes to the paper and it has been improved. I think the paper is nearly ready for publication but I have a couple of minor revisions it would be worthwhile to address:

1. Great that geology has been added to Figure 1 but it is not clear what the different colours mean (e.g. 'ev', 'it', 'mt'). The legend needs to be made clearer to the reader (or they need to be described in the caption).

We added the description to the caption.

2. Section 4.1 L275-286 in tracked change version of manuscript. While it is helpful to describe the main geologic and land cover of the different basins, this doesn't really fit in the results section at the moment. Can you link this back with the results in Figure 2?

We agree with the reviewer that the description of land cover and geology characteristics per cluster may not fully fit in the Results section as it is. To overcome this, we moved the paragraph describing geologic characteristics and linked it with the scatter in Fig. 2d. (L261-264 in the revised manuscript). The land cover characteristics are used later in the manuscript to interpret our results, in particular, when we discuss the potential anthropic influence on drought propagation (Discussion section, L543-L547).

Just to note that in future, it would be helpful if the authors could relate their responses and details of changes made to the paper to specific line/figure numbers in the track change or final document.

We'll do that.

[revised manuscript text omitted]